Manuscript prepared for Biogeosciences
with version 2015/09/17 7.94 Copernicus papers of the LATEX class copernicus.cls.
Date: 6 March 2018

# Modeling the biogeochemical impact of atmospheric phosphate deposition from desert dust and combustion sources to the Mediterranean Sea

Camille Richon[1,2], Jean-Claude Dutay[1], François Dulac[1], Rong Wang[1,3], and Yves Balkanski[1]

[1]LSCE/IPSL, Laboratoire des Sciences du Climat et de l'Environnement, CEA-CNRS-UVSQ, Gif-sur-Yvette, France
[2]Now at: Department of Earth, Ocean and Ecological Sciences, School of Environmental Sciences, University of Liverpool, Liverpool L69 3GP, UK
[3]Now at: Department of Environmental Science and Engineering, Fudan University, Shanghai 200433, China

*Correspondence to:* Camille Richon (camille.richon@lsce.ipsl.fr)

**Abstract.** Daily modeled fields of phosphate deposition from natural dust, anthropogenic combustion and wildfires were used to assess the effect of this external nutrient on marine biogeochemistry. The ocean model used is a high resolution (1/12°) regional coupled dynamical-–biogeochemical model of the Mediterranean Sea (NEMOMED12/PISCES). The input fields of phosphorus are for

2005, which is the only available daily resolved deposition fields from the global atmospheric chemical transport model LMDz–INCA. Traditionally, dust has been suggested to be the main atmospheric source of phosphorus, but the LMDz–INCA model suggests that combustion is dominant over natural dust as an atmospheric source of phosphate ($PO_4$, the bioavailable form of phosphorus in seawater) for the Mediterranean Sea. According to the atmospheric transport model, phosphate

deposition from combustion (*Pcomb*) brings on average 40.5 $10^{-6}$ mol $PO_4$ m$^{-2}$ year$^{-1}$ over the entire Mediterranean Sea for the year 2005 and is the primary source over the northern part (e.g., 101 $10^{-6}$ mol $PO_4$ m$^{-2}$ year$^{-1}$ from combustion deposited in 2005 over the North Adriatic against 12.4 $10^{-6}$ from dust). Lithogenic dust brings 17.2 $10^{-6}$ mol $PO_4$ m$^{-2}$ year$^{-1}$ on average over the Mediterranean Sea in 2005 and is the primary source of atmospheric phosphate to the southern

Mediterranean basin in our simulations (e.g., 31.8 $10^{-6}$ mol $PO_4$ m$^{-2}$ year$^{-1}$ from dust deposited in 2005 on average over the South Ionian basin against 12.4 $10^{-6}$ from combustion). The evaluation of monthly averaged deposition fluxes variability of *Pdust* and *Pcomb* for the 1997–2012 period indicates that these conclusion may hold true for different years. We examine separately the two atmospheric phosphate sources and their respective fluxes variability and evaluate their impacts on

marine surface biogeochemistry (phosphate concentration, chlorophyll *a*, primary production). The impacts of the different phosphate deposition sources on the biogeochemistry of the Mediterranean are found localized, seasonally varying and small, but yet statistically significant. Differences in

the geographical deposition patterns between phosphate from dust and from combustion will cause contrasted and significant changes in the biogeochemistry of the basin. We contrast the effects of
combustion in the northern basin (*Pcomb* deposition effects are found 10 times more important in the northern Adriatic, close to the main source region) to the effects of dust in the southern basin. These different phosphorus sources should therefore be accounted for in modeling studies.

## 1   Introduction

Atmospheric deposition is an important source for bioavailable nutrients to the remote oceanic wa-
ters (e.g. Jickells, 2005; Mahowald et al., 2009). Aerosols not only include nutritive elements such as nitrogen and phosphorus, which are the main limiting nutrients for marine primary productivity, but also trace metals (Dulac et al., 1989; Heimburger et al., 2012), among which copper has toxic effects on some phytoplankton species (Paytan et al., 2009). Aerosols can even be associated with living organisms such as viruses, fungae and bacteria (Mayol et al., 2014). The most important aerosol mass
deposition fluxes to the global ocean are induced by sea salt and natural desert dust (Goudie, 2006; Albani et al., 2015) respectively corresponding to material recycling and external inputs. In terms of nutrient fluxes, silica, nitrogen, iron and phosphorus are most abundant among the deposited nutrients (Guerzoni et al., 1999). Nitrogen, phosphate and iron are the three most important deposited elements measured in the Gulf of Aqaba, which is under the influence of both natural and anthro-
pogenic aerosols (Chen et al., 2007). It is especially important to better characterize external sources of phosphorus because it limits productivity (either as a primary, or secondary limiting nutrient) in many regions of the oceans (Moore et al., 2013). The main sources of atmospheric phosphorus for the surface waters of the global ocean are desert dust, and combustion from anthropogenic activities (Graham and Duce, 1979; Mahowald et al., 2008).
45       The Mediterranean Sea is highly oligotrophic and the intense summer vertical stratification leads to rapid nutrient depletion in surface waters (Bosc et al., 2004). During that season, the atmosphere is the only nutrient source for most of the Mediterranean surface waters (Markaki et al., 2003). Many studies discuss the impacts of atmospheric nutrient deposition to the oligotrophic Mediterranean Sea surface waters (Guerzoni et al., 1999; Markaki et al., 2003; Gallisai et al., 2014). Monitoring and
experimental studies have shown that deposition of great amounts of aerosols significantly impacts surface biogeochemistry over this basin (see Herut et al., 2005; Guieu et al., 2014). Ridame et al. (2014) showed that extreme events of Saharan dust deposition can double primary production and chlorophyll *a* concentration. In particular, the soluble fraction of this aerosol provides the main limiting nutrient to the Mediterranean: inorganic phosphorus (Bergametti et al., 1992; Krom et al.,
2010; Tanaka et al., 2011).
    Until now, Saharan dust was believed to be the most important atmospheric source of nutrients to the oligotrophic Mediterranean (e.g., Guerzoni et al., 1999). But the Mediterranean region is one

of the most densely populated areas of the world and many of the surrounding countries historically developed their capital cities along its coasts. The recent development of many of the Mediterranean countries has led to high anthropogenic footprint over ecosystems and climate through increased population and industrial activities emitting aerosols (Kanakidou et al., 2011). Moreover, aerosols are deposited to the Mediterranean Sea from a variety of different geographical sources. The impacts of aerosol deposition on the Mediterranean region are not fully understood and they may change in the future as a result of climate change impacts on land and sea (e.g. Peñuelas et al., 2013). The Sahara and Middle East are important sources of natural lithogenic dust (e.g., Ganor and Mamane, 1982; Bergametti et al., 1989; Al-Momani et al., 1995; Vincent et al., 2016) whereas the surrounding cities and highly industrialized areas are sources of atmospheric pollutants emitted by biofuels for heating and fossil fuel burning (Migon et al., 2001; Piazzola et al., 2016). The 85 million hectares of forests around the basin associated to the Mediterranean dry summer climate are also an occasional intense aerosol source due to wildfire emissions (Kaskaoutis et al., 2011; Poupkou et al., 2014; Turquety et al., 2014), providing for instance soluble iron to the Mediterranean (Guieu et al., 2005).

Modeling represents an interesting approach to investigate the impact of atmospheric nutrient deposition on oceanic biogeochemical cycles. Richon et al. (2017) use a regional coupled dynamical––biogeochemical high resolution model of the Mediterranean Sea to study the impacts of N deposition from natural and anthropogenic sources and phosphate from dust on the biogeochemistry of the Mediterranean Sea. Their results showed important impacts of N deposition on biological productivity (primary production, chlorophyll *a* production, plankton and bacterial concentrations) in the northern Ionian and Levantine basins and limited, yet significant impact of P deposition in the southern Mediterranean regions. In the present study, we extend this investigation of phosphate deposition effects ; by further considering the contribution of P from combustion sources in addition to that from anthropogenic activities and wildfires, and comparing the effects of desert dust and combustion inputs of phosphate on the marine surface nutrient and biogeochemical budgets.

## 2  Methods

We use mass deposition outputs from the global atmospheric model LMDz–INCA (Hauglustaine et al., 2014; Wang et al., 2015a) as external sources of phosphate in the regional high resolution coupled NEMOMED12/PISCES model. We consider separately phosphorus from desert dust and from combustion in order to isolate their respective effects as nutrient sources.

### 2.1  The oceanic model

We use the regional oceanic model NEMO (Madec, 2008) at a high spatial resolution of 1/12° over the Mediterranean (MED12). The 1/12° grid resolution is stretched in latitude and ranges between 6 km at 46° N and 8 km at 30° N. This fine–scale resolution enables us to represent important features

of the Mediterranean circulation that are small eddies. This grid has 75 unevenly spaced vertical layers, with depth ranging from 1 to 134 m from the surface to the bottom, and 10 levels in the first 100 m. The oceanic domain covers all the Mediterranean and a part of the Atlantic between

the Strait of Gibraltar and 11° W called the buffer zone. The regional NEMO–MED dynamical simulation used in this study to force the PISCES model is NM12–FREE, evaluated in Hamon et al. (2015). Atmospheric forcing conditions are prescribed from the ALDERA dataset (Hamon et al., 2015). Temperature and salinity are relaxed monthly to climatologies in the buffer zone (Fichaut et al., 2003). This simulation reproduces well the general circulation and variability of the wa-

ter masses characteristics. However, Hamon et al. (2015) identify some shortcomings: transports through the Strait of Gibraltar are underestimated by about 0.1 Sv, the circulation and mesoscale activity in the western basin (Algerian current, Northern current) are underestimated, and positive temporal drifts in the heat and salt content occur in the intermediate layer all over the Mediterranean Sea. This may lead to overestimation in temperature and salinity in intermediate waters after long

simulation periods (hundreds of years). The ability of the model to reproduce the general circulation of the water masses was also evaluated in a similar configuration with CFC (Palmiéri et al., 2015), neodymium (Ayache et al., 2016a), tritium–helium–3 (Ayache et al., 2015) and $^{14}C$ (Ayache et al., 2016b). These evaluations showed that the NEMO model is able to produce satisfying results when studying characteristics such as age–tracer of water masses or passive tracer transport.

The biogeochemical model PISCES (Aumont et al., 2015) is coupled to the physical model. The regional coupled configuration NEMOMED12/PISCES was developed by Palmiéri (2014) and Richon et al. (2017). Only two trophic levels are explicitly represented in PISCES: phytoplankton (autotrophic) and zooplankton (heterotrophic). Each plankton type is composed of two size classes: nanophytoplankton and diatoms; microzooplankton and mesozooplankton. PISCES is a Redfieldian

model: the C:N:P ratio used for plankton growth is fixed to 122:16:1. In PISCES, nutrient uptake (nitrate, ammonium, silicate, iron and phosphate) is governed by a Monod–type model (Monod, 1958). The concentration of nutrients is linked to phytoplankton productivity and chlorophyll $a$ production according to the equations described in Aumont et al. (2015). Phytoplankton growth rate is dependent on nutrient concentrations via the growth limiting factors (see Aumont et al., 2015, for

detailed equations). We prescribe riverine nutrients input fluxes from the estimation of Ludwig et al. (2009) that accounts for the nutrient fluxes from 239 rivers around the Mediterranean and Black Sea obtained from measurements and model data. The estimations of riverine fluxes are not available after 2000. Therefore, we use the riverine fluxes from the year 2000 in our study. Nutrient concentrations in the buffer zone are relaxed to the monthly climatology of the World Ocean Atlas (WOA)

(Locarnini et al., 2006). Nutrient fluxes from the Atlantic are computed in the model as the product of the nutrient concentrations in the buffer zone times the water fluxes through the Strait of Gibraltar.

The model is run in off–line mode like in the studies performed by Palmiéri et al. (2015), Guyennon et al. (2015), Ayache et al. (2015, 2016a, b) and Richon et al. (2017). PISCES biogeochemical

tracers are transported using an advection–diffusion scheme driven by dynamical variables (veloc-
ities, pressure, mixing coefficients...) previously calculated by the oceanic model NEMO. Biogeo-
chemical characteristics of the latest version of the NEMOMED12/PISCES model are evaluated in
Richon et al. (2017). In particular, NEMOMED12/PISCES produces well the West–to–East gradient
of productivity when compared to satellite chlorophyll *a* estimates, and simulates the main produc-
tive zones located in the Alboran Sea, the Gulf of Lions and most coastal areas but with a lower
amplitude (see Figure 1 and Table 1). We filtered out chlorophyll *a* values in coastal areas (Bosc
et al., 2004). The vertical distribution of nutrients is globally satisfyingly simulated, despite a too
sharp nutricline in the intermediate waters in some sub–basins (e.g. Alboran, South Ionian).

Additional statistical indicators provided in Appendix A show a good reproduction of the nutri-
ents and the chlorophyll *a* vertical concentrations in different Mediterranean regions. However, the
model fails to reproduce surface and intermediate phosphate concentrations in some regions such as
the South Levantine. It is important to note that the nutrient concentrations in surface waters, espe-
cially phosphate concentrations in the ultra–oligotrophic eastern basin, are often below the detection
limit of measuring devices. Therefore, the negative bias in model surface concentration estimates
may be linked with measurement uncertainties. Richon et al. (2017) calculated the average and stan-
dard deviation of chlorophyll *a* values measured and modeled at the DYFAMED station (Ligurian
Sea) for the 1997–2005 period and found that the average measured chlorophyll *a* in the top 200
meters is $0.290 \pm 0.177 \ 10^{-3}$ g m$^{-3}$ and the average model value is $0.205 \pm 0.111 \ 10^{-3}$ g m$^{-3}$.
For PO$_4$, the average measured value is $0.234 \pm 0.085 \ 10^{-3}$ mol m$^{-3}$ and the modeled average is
$0.167 \pm 0.179 \ 10^{-3}$ mol m$^{-3}$. We report in Appendix A additional figures on the comparison of
modeled and measured PO$_4$ and NO$_3$ concentrations. Despite some unavoidable shortcomings, the
performances of the model are reasonable for conducting our scientific investigation.

## 2.2   Atmospheric deposition of phosphate

The objective of this study is to use consistent atmospheric phosphate inputs from contrasted sources
simulated by the same atmospheric model. Hence, we selected daily atmospheric deposition fields
of total phosphorus (P) from natural dust and combustion both simulated with the LMDz–INCA
chemistry–climate global model (Wang et al., 2015b). This global model has a rather low spatial
resolution of 0.94° in latitude × 1.28° in longitude. The daily deposition fluxes from these two
sources have been simulated globally solely for a one year period (2005). The form of deposited
phosphorus in the model is the same for all sources and is considered to be soluble phosphate (PO$_4^{3-}$)
which is the bioavailable form of phosphorus in PISCES (Aumont and Bopp, 2006). We considered
that given the high spatial and temporal variability of atmospheric deposition fluxes, a monthly
resolution of deposition, as available for other years (Wang et al., 2017), would be a too strong and
unnecessary limitation in simulating the biogeochemical response.

In this study, we first include natural desert dust as a source of phosphorus. In the atmospheric model, desert dust emissions are computed every 6 hours using the European Centre for Medium-Range Weather Forecasts (ECMWF) wind data interpolated to the LMDz grid. Following Mahowald et al. (2008), we consider that only 10 % of phosphorus from desert dust is bioavailable PO$_4$ (hereafter named *Pdust*). This solubility value has also been used by other authors (see also Anderson et al., 2010). Izquierdo et al. (2012) report an average solubility of phosphorus of about 11 % in African dust–loaded rains in North–East Spain. In addition, the daily total dust deposition simulated with similar forcings on a 1.27° in latitude by 2.5° in longitude grid is available for the period 1997–2012. We used this time series of total dust deposition to compare the year 2005 with the average inter–annual deposition flux. We found that the yearly average deposition of dust from 2005 is close to the multi–year depositional average (not shown).

The second source of atmospheric phosphorus is combustion. Here, the term combustion entails anthropogenic combustion from energy production, biofuels, and wildfires emissions (hereafter named *Pcomb*). The *Pcomb* deposition fields used here were obtained from a different simulation performed with the same model LMDz–INCA on a coarser grid resolution than for *Pdust* of 1.27° in latitude by 2.5° in longitude. In the atmospheric model, phosphorus emissions from combustion due to anthropogenic activities are assumed constant throughout the year 2005, only wildfire emissions vary monthly based on the GFED 4.1 data set for biomass burning (van der Werf et al., 2010). According to this data set, wildfire emissions around the Mediterranean for 2005 are close to the inter–annual average for the period 1997–2009. Wang et al. (2015a) estimated global *Pcomb* emissions based on the consumption of different fuels (including wildfires) and the P content in all types of fuels for more than 222 countries and territories. We consider that 54 % of the total emitted P from combustion is bioavailable phosphate (*Pcomb*) (Longo et al., 2014). Up to now, there are at least two major model–based data sets of *Pcomb* deposition. Mahowald et al. (2008) had published the first map of P deposition from combustion sources based on a bottom–up P emission inventory, which leads to a general agreement of their globally modeled surface P concentrations with measurements with a large underestimation in the modeled P deposition. Wang et al. (2015a, 2017) revised the emission factors from all fuel types burned during combustion to come up with a new global inventory of phosphorus emitted from coal, biofuels and biomass burning. This inventory amounts to an emission of P from these 3 sources of 1.96 Tg P yr$^{-1}$ for the year 1996 (Wang et al., 2015a), which is 28 times the inventory of 0.070 Tg P yr$^{-1}$ compiled by Mahowald et al. (2008) for the same year. The same authors (Wang et al., 2015a) evaluated both P surface concentrations and P deposition from LMDz–INCA and showed no systematic bias against measurements taken at the global scale. In general, the LMDz–INCA modeled atmospheric P deposition fluxes have been evaluated globally by comparing time series of deposition measurements, showing a significantly reduced model bias relative to observations when considering the contribution of P emissions from combustion than

when considering only P from dust (Figure 4 in Wang et al., 2015a). However, it should be noted that there were only three sites with time series of P deposition over the Mediterranean region.

In order to assess inter–annual variability of the *Pcomb* deposition, Figure 2 shows the annual average *Pcomb* deposition over the Mediterranean for the 1997–2012 period. Figure 3 shows the standard deviation to average ratio of *Pcomb* deposition over the Mediterranean computed from LMDz–INCA for the 1997–2012 period. These figures show that *Pcomb* deposition has low inter–annual variability. Therefore, 2005 can be considered as a study year because both *Pcomb* and *Pdust* deposition are close to the inter–annual average for the 1997–2012 period.

Another important input of P aerosols in this region is from sea spray (Querol et al., 2009; Grythe et al., 2014; Schwier et al., 2015). Sea spray aerosols over the Mediterranean mainly come from the Mediterranean itself with little contribution from the Atlantic Ocean. Therefore, the net contribution of P from sea spray is considered negligible in our simulations.

Active volcanoes around the Mediterranean such as the Etna or the Stromboli are another potential source for aerosols. Phosphorus mass in volcanic aerosols is very low (P. Allard, pers. comm.) although it is considered to be almost entirely soluble (Mahowald et al., 2008). Finally, the 85 million hectares of forest around the Mediterranean are a potential source of biogenic particles such as pollen and vegetal debris (Minero et al., 1998) that contain phosphorus. The total mass flux of phosphorus from biogenic particles seems to be important on the global atmospheric phosphorus budget (Wang et al., 2015a). It is not included in our study, which can be seen as a potential limitation. However, biogenic particles have very low solubility in seawater (Mahowald et al., 2008). The LMDz–INCA model provides the summed bulk deposition of both phosphorus from volcanoes and biogenic particles (named PBAP). We chose to discard PBAP as a source of P since these 2 contrasted sources have very different solubilities but cannot be apportioned within PBAP.

## 2.3 Simulation set–up

We ran NEMOMED12/PISCES for one year with the 2005 physical and biogeochemical forcings. Initial conditions at the end of 2004 are taken from the 1997–2012 simulation described in Richon et al. (2017) including anthropogenic nitrogen deposition ("N" simulation). The reference simulation (REF) is a simulation performed with no atmospheric deposition of phosphate as described in Richon et al. (2017).

We investigate the impacts of each source of $PO_4$ by performing two different simulations: "PDUST" and "PCOMB"; they include, respectively, natural dust only and combustion–generated aerosol only as atmospheric sources of $PO_4$. We also performed a "Total P" simulation with the two sources included. From now on, we use "total P" to indicate the sum of bioavailable phosphate from dust and combustion (*Pdust + Pcomb*). The results presented in this study are based on the relative differences between the simulations. For instance, the impacts of *Pdust* are calculated as the difference between PDUST and REF simulations (PDUST-REF).

## 3 Results

### 3.1 Evaluation of P deposition fluxes

Very few measurements of atmospheric phosphorus deposition exist over the Mediterranean region. Moreover, it is difficult to apportion between different sources when analyzing bulk deposition samples. We did not find any available time series of total phosphorus deposition in the Mediterranean covering our simulation period. Therefore, we compare the monthly P deposition flux from LMDz–INCA with the non time–consistent monthly fluxes over years as close as possible to 2005. Estimates of Turquety et al. (2014) indicate that 2005 is not an exceptional year for fires and the time series of natural dust deposition modeled with LMDz–INCA indicate that the deposition flux of 2005 is close to the inter–annual average (not shown). We used the time series of phosphorus measured at 9 different stations over the Mediterranean from the ADIOS campaign (Guieu et al., 2010) and the soluble phosphate from deposition measurements at 2 stations in the South of France from the MOOSE campaign (de Fommervault et al., 2015). The ADIOS time series cover June 2001 to May 2002 and the sampling sites cover almost all regions of the basin. The MOOSE time series cover 2007 to 2012. We use the time series of average monthly flux in $10^{-6}$ g $PO_4$ $m^{-2}$ month$^{-1}$ and compare it with our model average monthly fluxes in the grid cells corresponding to the stations. Figure 4 shows the comparison between modeled and measured fluxes in terms of geometric means and standard deviations of monthly values of each time series. The fluxes are highly variable according to the station and the season (variability spans over several orders of magnitudes). Our comparison must be taken with caution since we compare different years in the model and the observations.

We were able to compare the dust deposition flux modeled with LMDz–INCA used to derive *Pdust* deposition over the ADIOS sampling period with the measurements. The comparison is shown in Table 2. The dust fluxes produced by the model at several stations are realistic, in spite of the low spatial variability from the dust fluxes produced by the global model LMDz–INCA even though the geometric standard deviations of the fluxes can be regionally very high. In Table 2 the dust deposition fluxes for the period 2001–2002, corresponding to the ADIOS campaign, are based on model outputs with a lower resolution (1.27° in latitude by 2.5° in longitude) than those for the year 2005 (0.94° in latitude by 1.28° in longitude). As stated by Bouet et al. (2012), dust emission (and hence its deposition) is highly sensitive to model resolution. The coarse resolution of LMDz may significantly reduce the total dust emission in the model but also reduce surface winds and aerosol transport (see Discussion section hereafter). Therefore, the coarse resolution of the dust model used in Table 2 for 2001–2002 may explain the underestimation and the lack of spatial variability from the model. We also noted a better agreement (within a factor of 2) at the 4 stations of the eastern Mediterranean (Cyprus, Greek Islands and Turkey).

In order to assess properly the performance of the atmospheric model in reproducing deposition fluxes, we would need continuous times series of deposition in different stations over the Mediter-

ranean and simulations covering the measurement periods. Our comparison is at the moment the most feasible evaluation with the existing data over the Mediterranean. This diagnostic reveals that the model probably tends to underestimate the P deposition from both dust and anthropogenic sources. These results are consistent with Wang et al. (2015a). The underestimation of total P deposition is also likely due in part to our omission of P from other potential sources such as PBAP and sea salt.

## 3.2 Characterization of phosphate deposition from the different sources

The 2005 seasonal spatial distribution of *Pdust* deposition is shown in Figure 5. *Pdust* deposition is highly variable in space and time. It is maximal in spring (MAM). In this season, the main dust source is the Sahara and it affects mostly the eastern basin (Moulin et al., 1998). In winter (DJF), the influence of dust from the Middle East is observed (Basart et al., 2012). In summer (JJA) and autumn (SON), the deposition is at its minimum and located close to the southern Ionian coasts. Average deposition flux over the basin is $0.122 \cdot 10^9$ g $PO_4$ month$^{-1}$ with notable monthly variability (standard deviation = $0.102 \cdot 10^9$ g $PO_4$ month$^{-1}$). This seasonal cycle of dust deposition is similar to the one simulated by the regional model ALADIN–Climat (Nabat et al., 2012) used in Richon et al. (2017) but LMDz *Pdust* deposition flux is significantly lower than that from ALADIN (see Discussion section).

The seasonal spatial distribution of *Pcomb* deposition is shown in Figure 6. Atmospheric deposition of phosphate from combustion is on average $0.258 \cdot 10^9$ g $PO_4$ month$^{-1}$ over the entire basin. It amounts to twice the atmospheric deposition of phosphate from desert dust ($0.122 \cdot 10^9$ g $PO_4$ month$^{-1}$). The seasonal variability of *Pcomb* deposition is lower than for *Pdust* (standard deviation of *Pcomb* = $0.046 \cdot 10^9$ g $PO_4$ month$^{-1}$). This is linked to the anthropogenic nature of *Pcomb* emissions and the low contribution of atmospheric transport to seasonal variability. Maximal deposition occurs in summer, likely due to the forest fires around the Mediterranean. In particular, we observe higher deposition close to the Algerian, Spanish and Italian coasts in summer. These countries are particularly subject to dry and hot summer conditions that favor forest fires (Turquety et al., 2014). We observe a high spatial variability in the deposition field with a North–to–South decreasing gradient in deposition, the major part of total mass being deposited close to the coasts, especially in the Aegean Sea. The presence of many industrial areas around the Adriatic and Aegean explains the high deposition fluxes observed in these regions. In the Aegean Sea, *Pcomb* deposition constitutes a more than 4 times greater phosphate source than desert dust (respectively $0.0529 \cdot 10^9$ g $PO_4$ month$^{-1}$ and $0.0118 \cdot 10^9$ g $PO_4$ month$^{-1}$ for *Pcomb* and *Pdust* average deposition). According to our model forcings, for which uncertainties are still large, the riverine inputs would constitute the main phosphate source to the Mediterranean Basin ($3.16 \cdot 10^9$ g $PO_4$ month$^{-1}$ at the basin scale). These inputs alone account for over 85 % of the total (atmospheric + riverine input + Gibraltar Strait) as documented in Table 3.

Figure 7a illustrates that combustion is the dominant source of atmospheric phosphate to the Mediterranean basin for the year 2005. This map shows the average proportion of *Pdust* in total phosphorus deposition (*Pdust+Pcomb*). The results indicate that *Pdust* accounts for 30 % of phosphorus deposition on average in 2005 at the basin scale, and is only dominant along the southern Mediterranean coast, in the Gulf of Libya. This map highlights the contrasted areas influenced by the different atmospheric P sources. The North of the basin is primarily under the influence of combustion aerosol sources, and the South of the basin is under the influence of dust aerosol sources.

Evaluation of deposition fluxes for the period 1997–2012 (for which monthly deposition fluxes are available) shows a continuous dominance of *Pcomb* fluxes at the basin scale (see Figure 7b). These results agree with the ones of Desboeufs et al. (in prep) who noted that combustion aerosols are responsible for 85 % of P deposition in the northwestern coast of Corsica over a three years period (2008–2011, versus 15 % of P from dust).

Figure 8 shows the contrasted contribution of the respective fluxes for the month of June 2005. Our previous study showed that June is the period of most important impacts from aerosol deposition on surface marine productivity in spite of the low fluxes, due to thermal stratification (Richon et al., 2017). The relative contribution of *Pdust* and *Pcomb* deposition fluxes are compared over three regions: the North Adriatic, the South Adriatic and the South Ionian (See Figure 5). We defined these regions as in Manca et al. (2004). They were selected as they highlight three contrasted conditions. The North Adriatic is under strong influence of both riverine inputs and atmospheric deposition of P from combustion (Figure 8), the South Adriatic encompasses atmospheric coastal deposition but is distant from major riverine inputs, and the South Ionian is a deep, highly oligotrophic area. The deposition flux of *Pcomb* is maximal in the northern Adriatic. In this basin, *Pcomb* flux is five times higher than *Pdust*. However, *Pdust* deposition flux increases towards the South to reach a value three times higher than *Pcomb* flux upon reaching the southern Ionian coasts. This spatial distribution of deposition is also found by Myriokefalitakis et al. (2016).

By including different sources of atmospheric phosphate from the same model, we can compare the relative contribution of each atmospheric source with the other external nutrient suppliers (rivers and Gibraltar). Table 3 shows the relative contribution simulated by the model of atmospheric phosphate sources in total external phosphate supply to the Mediterranean basins. Our estimations of total aerosol contribution to $PO_4$ supply are slightly lower than the literature values. Table 3 shows that, in the model, *Pcomb* is dominant over *Pdust* as a source of atmospheric phosphate at the basin scale for the year 2005. This dominance is found in all regions of the Mediterranean, except in the Ionian Sea where *Pdust* and *Pcomb* contributions are equivalent. We note that the estimates from Krom et al. (2010) were calculated by extrapolating to the eastern basin measurements from very few locations in Turkey and Greece. Vincent et al. (2016) report that recent desert dust deposition fluxes have decreased in the 2010s by an order of magnitude compared to the 1980s that Krom et al. (2010) refer in part to. This may explain that we find combustion to be a more important source

of atmospheric phosphate at the basin scale in 2005 in comparison to natural dust. In the pelagic Ionian basin, *Pdust* and *Pcomb* contributions are comparable on a yearly average (20 %). However, combustion represents at most a third of the contribution whereas dust–derived phosphate deposition is more seasonally variable and can be the major source of $PO_4$ for this basin during spring (contribution of *Pdust* to $PO_4$ supply up to 60 %).

## 3.3 Impacts of atmospheric deposition on marine surface phosphate

Atmospheric deposition of phosphate aerosol has different impacts in the model on $PO_4$ concentration depending on the source, the location, and the period of the year. The impacts of deposition depend on the flux and the underlying biogeochemical conditions in the water column. Even though the deposition fluxes are very low during the stratified season, their relative impacts are maximal

because the major part of the Mediterranean is highly limited in nutrients (Richon et al., 2017).

Figure 9 shows the relative impacts of phosphate deposition from the two sources (combustion and dust) on surface $PO_4$ concentration for the month of June 2005. We can distinguish 3 different responses to nutrient deposition: two non responsive zones that are either not nutrient limited or limited in more than one nutrient and a responsive zone limited in the deposited nutrient. In the

regions under riverine input influence such as the North Adriatic, relative impacts of atmospheric deposition are low even though the fluxes of *Pcomb* are maximal because the Pô river delivers high amounts of nutrients in this area. In very unproductive regions such as the South Ionian basin, we observe very low impacts of deposition on $PO_4$ concentrations (between 5 and 12 % enhancement close to the Libyan coast). This basin is highly depleted in nutrients, especially in summer. But the

deposition fluxes are very low ($90 \ 10^{-6}$ mol m$^{-2}$ of total $PO_4$). This low fluxes of nutrients are probably consumed very fast and do not yield a strong concentration enhancement. Finally, some areas respond strongly to phosphate deposition. We observe $PO_4$ surface enhancement over 40 % in the South Adriatic, Tyrrhenian and North Aegean basins. These regions are under some nutrient sources influence; they are not fully pelagic and receive nutrients from coasts or upwelling (Sicily

Strait front). The high response to phosphate deposition indicates that these regions are primarily P–depleted.

These contrasted results indicate that the relative impacts of atmospheric deposition from different sources are dependent on both the underlying phosphate concentration and the bioavailable phosphate deposition flux. The relative biogeochemical impacts of $PO_4$ deposition are variable due

to the biogeochemical state of the region.

## 3.4 Biogeochemical impacts of P deposition

In the PISCES model, atmospheric deposition of nutrient is treated as an external forcing. The effects of the different aerosols on the Mediterranean biogeochemistry are considered simply additive. Fluxes of nutrients are added to the total pool of dissolved nutrients according to their deposition

flux and chemical properties (fixed solubility and chemical composition). The effects of total P de-
position on marine biogeochemistry are a combination of effects of the two P sources in this model
version (Table 3).

We focus here on the month of June that shows maximum impacts of deposition because of sur-
face water stratification. Figure 10 shows the average relative effects of P deposition on surface
chlorophyll *a*. The relative effects of total P deposition on surface chlorophyll *a* concentration are
modest. The majority of *Pdust* effects on surface chlorophyll *a* are in the southwestern basin along
the Algerian coasts. *Pcomb* has maximal impacts in the North of the basin, in areas of high depo-
sition. However, *Pcomb* also affects the area influenced by *Pdust* in the South. In this Redfieldian
version of PISCES, chlorophyll *a* production is linked with nutrient uptake that is constrained by the
Redfield ratio. Therefore, the addition of excess nutrient will enhance chlorophyll *a* production as
long as other nutrients are bioavailable in the Redfield proportions.

Figure 11 shows the relative impacts of phosphate deposition from the two sources on surface
total primary productivity for the month of June. We observe that combustion—derived phosphate
has the greatest impacts on surface biological production: averaged regionally over the framed areas,
the enhancement in daily primary productivity is between 1 and 10 % but local maxima are up to
30 %.

The effects of atmospheric phosphate deposition are variable according to the source type. As for
chlorophyll *a*, dust–derived phosphate deposition has maximal impacts in the southern part of the
basin close to the Algerian and Tunisian coasts. The relative impacts of *Pdust* deposition in the South
Ionian basin are very low (about 1.7 %). This region of the Mediterranean is highly oligotrophic and
lacks all major nutrients, especially in summer. Nutrient co–limitations associated to minimal *Pdust*
deposition flux in summer explain the weak impacts of phosphate deposition in this area.

*Pcomb* deposition has maximal impacts in the North of the basin in the Adriatic and Aegean Seas.
In the northern Adriatic, the relative impacts of *Pcomb* deposition are lower than in the southern
Adriatic because the proximity of riverine inputs in the North reduces the relative importance of
atmospheric deposition in nutrient supply. The North of the Adriatic is generally productive all year
long. We can identify in Figure 11 the area in the Adriatic influenced by riverine inputs. This zone
encompasses the North of the Adriatic and the western coast down to the region of Bari (40°N,
17°E). In this area, atmospheric deposition has low influence on primary productivity (below 15 %).
This is in contrast with the southeastern part of the Adriatic under low riverine input influence. There,
we observe high impacts of atmospheric deposition and especially of *Pcomb* on primary productivity
(11 % on average but up to 50 % daily enhancement at some points). *Pcomb* and *Pdust* have similar
influences over the South of the basin.

In general, we can identify 3 different biogeochemical responses in the 3 areas indicated in Fig-
ure 11. Our hypothesis is that the different responses are linked to nutrient limitations. In the North
Adriatic, the influence of coastal nutrient inputs leads to low nutrient limitation and high productiv-

ity. In the South Adriatic, the high impact of atmospheric phosphate deposition may be the sign of important phosphate limitation. Finally, the lack of response in South Ionian in spite of the atmospheric phosphate deposition probably indicates that the region is co–limited in P and N.

As for the effects on $PO_4$ concentration, we observe different impacts of P deposition on primary production according to the nutrient status of the region. We find very low deposition impacts in nutrient repleted areas (e.g. North Adriatic), very low to no response in highly nutrient limited areas such as the South Ionian, and high response in areas limited by phosphorus only (e.g. South Adriatic).

## 4 Discussion

In contrast to the global ocean, combustion appears as an important source of atmospheric bioavailable phosphorus to the surface waters of the Mediterranean Sea due to the proximity of populated and forested areas. Based on our large scale LMDz–INCA model, we estimate that combustion is responsible for 7 % on average of total $PO_4$ supply. In comparison, the average contribution of *Pdust* to $PO_4$ supply is 4 % (Table 3). These estimates are based on our modeling values and take into

account only the sources of phosphate that are included in the simulations (namely rivers, Atlantic inputs, natural and combustion derived atmospheric phosphate). This provides an estimation of the relative importance of the 2 atmospheric sources under the specific conditions of the year 2005, but the restriction to only this particular year limits our conclusions. For this reason, the purpose of this study is to raise questions on the relative importance of the various aerosol sources that bor-

der the Mediterranean and their potential impacts on the nutrient supply and biological productivity of the basin. Saharan dust is a major source of particles in the Mediterranean (D'Almedia, 1986; Loÿe-Pilot et al., 1986) and does have an impact on the regional climate system (Nabat et al., 2012, 2015). The literature on Mediterranean aerosols is often centered on Saharan dust deposition, which is believed to have the largest impact on the basin's biogeochemistry (e.g. Bergametti et al., 1992;

Migon and Sandroni, 1999; Aghnatios et al., 2014). This study provides the first Mediterranean assessment of the contribution of another source of atmospheric phosphate than dust. It highlights that other sources, namely combustion, might be a dominant source of bioavailable phosphorus to Mediterranean surface waters.

The relative dominance of combustion over dust as a source of phosphate for the Mediterranean is

confirmed by the analysis of monthly modeled deposition fluxes of *Pdust* and *Pcomb* over the 1997–2012 period (see Figure 7). *Pcomb* dominates *Pdust* contribution to $PO_4$ supply over the northern basin (Adriatic and Aegean Seas in particular). For these regions in the vicinity of anthropogenic sources, *Pcomb* deposition has a low variability whereas *Pdust* deposition occurs during transient events and is therefore highly variable on a monthly basis. This was already noticed by Bergametti

et al. (1989) and Gkikas et al. (2016) who describe the majority of dust as occurring in a few episodic deposition events, whereas anthropogenic aerosols have a more constant flux. These results are also

coherent with Rea et al. (2015) who estimate anthropogenic emissions to be the main component of $PM_{2.5}$ (particulate matter with diameter $< 2.5$ $\mu$m) and dust to be the main component of $PM_{10}$ (particulate matter with diameter $< 10$ $\mu$m) over the Mediterranean. The maximal contribution of

atmospheric deposition to $PO_4$ budgets is observed in spring, when the deposition fluxes are maximal. In summer, the relative contribution in each sub–basin is very small because the flux of *Pdust* is very low. The high, nearly constant fluxes of *Pcomb* deposited close to the coasts, especially in semi-–closed sub-—basins such as the Adriatic and Aegean, constitute the major source of soluble atmospheric phosphate to the surface of the Mediterranean Sea. Although total mass deposition of

particulate phosphorus from desert dust exceeds that of combustion aerosols, the latter are much more soluble than lithogenic dust. This explains in our results the yearly predominance of *Pcomb* as a source of bioavailable phosphate. However, the underestimation of deposition fluxes indicated by Figure 4 limits the conclusions we can draw from our results on the relative contributions of the different phosphate external sources. More measurements and developments of the atmospheric model

must be undertaken to make more precise assessments of the importance of atmospheric deposition as a source of nutrients for the oligotrophic Mediterranean.

     The LMDz–INCA model version used in this study integrates constant emissions of *Pcomb* from anthropogenic sources. The variability of this deposition flux is only due to variability of atmospheric transport and deposition processes such as winds and rain or dry sedimentation. The atmospheric

model LMDz–INCA has a low resolution given the regional Mediterranean scale: *Pdust* deposition forcing has 280x193 grid points globally and $\sim$500 grid points covering the Mediterranean, and *Pcomb* forcing has 144x143 grid points in total and $\sim$200 grid points covering the Mediterranean. These forcings reproduce realistic deposition patterns at the global scale, in spite of generally underestimating the measured fluxes (Wang et al., 2017) but may not be reliable when analyzing small

scale deposition patterns. There is to our knowledge no regional model Mediterranean model available that represents phosphorus deposition from both natural and anthropogenic sources. Investigating these atmospheric deposition fluxes from a higher resolution regional model is a perspective to consider in order to strenghen our conclusions on the spatial distribution of *Pdust* and *Pcomb* influences.

Concerning the dust deposition component for which products from high resolution model exist (see the high resolution model ALADIN–Climat used in Richon et al., 2017), the overall average deposition estimation from the global model we use in this study appears much lower ($0.122 \pm 0.102$ $10^9$ g month$^{-1}$ over the Mediterranean in 2005 simulated with LMDz-–INCA and $0.568 \pm 0.322$ $10^9$ g month$^{-1}$ simulated with ALADIN-–Climat). Table 3 in Richon et al. (2017) shows the same comparison be-

tween measured dust fluxes and the dust fluxes from the ALADIN–Climate regional model than in Table 2. The fluxes reproduced by this 1/12° resolution regional model are generally closer to the measurements. The coarse resolution of LMDz may lead to a global underestimation of the dust emission fluxes, as shown by Bouet et al. (2012). Moreover, the higher spatial resolution of

ALADIN--Climat allows one to better reproduce intense regional winds (Lebeaupin Brossier et al.,
2011) that can favor transport of continental aerosols to the remote sea. Natural dust emissions,
transport and deposition to the Mediterranean are shown to be highly variable from a year to the next
(e.g. Moulin et al., 1997; Laurent et al., 2008; Vincent et al., 2016) so that the relative contributions
of *Pcomb* and *Pdust* may also vary. However, dust deposition fluxes available between 1997 and
2012 from the LMDz–INCA model indicate that 2005 is not an exceptional year (see also Figure 7).
Similarly, the inter–annual time series of dust deposition analyzed in Richon et al. (2017) showed
that 2005 is also not an exceptional year in the ALADIN–Climat model. The recent estimate of burnt
areas in the Euro–Mediterranean countries over 2003–2011 by Turquety et al. (2014) indicates a $\pm$
50 % annual variability, but it is impossible to separate anthropogenic and wildfires in *Pcomb* de-
position at present. Simulating separately atmospheric deposition from anthropogenic and wildfires
would give interesting perspectives on combustion aerosol deposition.

The reproduction of small scale atmospheric patterns such as coastal breezes that can transport
aerosols far from the coasts above the marine atmospheric boundary layer is also limited at the low
spatial resolution of LMDz (Ethé et al., 2002; Lebeaupin Brossier et al., 2011). This leads to low
day–to–day variability in total *Pcomb* deposition flux together with much larger modeled fluxes in
coastal areas. *Pcomb* deposition is limited in the model to coastal areas. However, our results in-
dicate that *Pcomb* is dominant over *Pdust* in this instance as an atmospheric source of phosphate
at the basin scale. Moreover, the atmospheric deposition model seems to underestimate phosphate
deposition in most of the stations we found (see Figure 4). Constant emissions of phosphate from
anthropogenic combustion is, however, a satisfying first approach because it permits to highlight the
high concentration contributed from industries and major urban centers around the Mediterranean.
However more refined emission scenarios would be interesting to consider in future modeling stud-
ies.

Some areas receive phosphate with different contributions from different sources (Figures 5, 6). In
particular, islands in the Eastern basin such as the Greek Islands, Crete and Cyprus receive phosphate
from the two sources, sometimes in a single deposition event (Koulouri et al., 2008; D'Alessandro
et al., 2013). Atmospheric processing of different aerosols will alter the nutrient composition and
solubility of this deposition (Migon and Sandroni, 1999; Desboeufs et al., 2001; Anderson et al.,
2010; Nenes et al., 2011; D'Alessandro et al., 2013). However our study does not account for such
mixing.

The atmospheric model used in our study does not provide biogenic and volcanic phosphorus
deposition separately. The model of Myriokefalitakis et al. (2016) allows to represent a more com-
plex atmospheric chemistry. This work showed that many different atmospheric P sources exist. In
particular, they estimate 0.195 and 0.006 TgP year$^{-1}$ of global emissions from biogenic and vol-
canic sources respectively. In the Mediterranean region that is surrounded by many forested areas,
biogenic emissions may be an important source of atmospheric phosphorus in the form of organic

matter. Moreover, Kanakidou et al. (2012) show that an important fraction of organic phosphorus can be emitted from combustion. In particular, the numerous forest fires occurring every summer in the Mediterranean region may constitute an important source of organic phosphorus. However, the PISCES version used in this study does not include organic phosphorus. In the ocean, organic phosphorus can be recycled by bacterial activity into inorganic phosphate that is bioavailable for plankton growth. Therefore, the inclusion of organic phosphorus in PISCES along with an estimation of organic phosphorus from atmospheric fluxes is a perspective to consider.

The PISCES version used in this study is based on the Redfield hypothesis that C:N:P ratios in organic cells are fixed. This fixed value determines the nutrient ratio for uptake and has the advantage of simplifying calculations in the 3–D high resolution coupled model and is supported by some observations (Pujo-Pay et al., 2011). However, because the Mediterranean is highly oligotrophic, this Refieldian hypothesis is questioned and the biogeochemical cycles may be determined by non–Redfieldian nutrient use. This non–Redfieldian behavior may imply complex nutrient limitations and co–limitations processes (Geider and La Roche, 2002) that can not be studied with the present PISCES version. As of today, there is no version of PISCES that includes the non–Redfieldian biogeochemistry in the Mediterranean. The development and use of such a version of PISCES is a perspective of this work that may help to fully understand nutrient dynamics and growth limitation process in the Mediterranean (Saito et al., 2008; Krom et al., 2010). However, this study provides interesting first results on the potential impacts of phosphate atmospheric deposition on the Mediterranean nutrient pool and potential implications on biological productivity. Moreover, the development and qualification of a non–Redfieldian version of the PISCES model may take several years. Plus, even if non–Redfieldian regional Mediterranean biogeochemical models such as ECO3M exist (Baklouti et al., 2006), their higher complexity leads also to a hard task, since the sensitivity of such models to parameter values is a delicate question that requires important computing time and data to solve before revisiting our conclusions.

## 5 Conclusions

This study is a first approach to quantify the effects of different atmospheric sources of phosphorus to the Mediterranean Sea surface. Our results indicate that contrary to the global ocean, combustion may be dominant over natural dust as an atmospheric source of phosphate for the Mediterranean Basin. This study is the first to examine separately the effects of atmospheric deposition of phosphate from different sources that have different seasonal cycles and deposition patterns over the Mediterranean Sea. According to our low resolution atmospheric model, phosphate deposition from combustion (which includes forest fires and anthropogenic activities) is mainly located close to the coasts and has low variability whereas phosphate deposition from dust is episodic and more widespread. The results indicate that combustion sources are dominant in the North of the basin

close to the emission sources whereas natural dust deposition is dominant in the South of the basin and is strongly dominant in pelagic areas such as the Middle Ionian and Levantine basins. The study of atmospheric model low resolution deposition fluxes over the period 1997–2012 indicate that the dominance of *Pcomb* over *Pdust* in the Mediterranean basin is consistently observed over this time period. The yearly–averaged deposition patterns are constant over the period. The relative effects of each source are maximal in their areas of maximal deposition and can induce an enhancement of up to 30 % in biological productivity in the top 10 meters during the period of surface water stratification.

In the coastal Adriatic and Aegean Seas that are under strong influence of anthropogenic emissions, we showed that combustion-derived phosphorus deposition may have effects on the biological productivity. It seems that only dust transported through large events can reach and fertilize pelagic waters. However, the pelagic zones far from coastal influence are often highly oligotrophic and co–limited in nutrients. Then, the deposition of one type of nutrient cannot relieve all the nutrient limitations to have strong fertilizing effect.

In spite of the limitations of our study linked to the availability of atmospheric P emission and the limited knowledge on atmospheric mixing processes impacts on bioavailability of deposited $PO_4$, we showed that atmospheric P deposition is an important source of bioavailable nutrients and has low but significant impacts on marine productivity. Combustion and soil dust sources display contrasted deposition patterns. Therefore, none should be neglected when accounting for atmospheric sources of nutrients in land and ocean biogeochemical models.

Our study highlights the difficulty to constrain atmospheric deposition in models because very few estimates of the deposition fluxes over the Mediterranean are available. The existing time series cover only very limited areas of the basin and short time periods. Plus, there is, to our knowledge, only one experimental study addressing the source apportionment of phosphate deposition. Longo et al. (2014) measured the solubility of P aerosols coming from South and North regions of the Mediterranean and showed that aerosols from Europe deliver more soluble P. Also, Desboeufs et al. (in prep) showed that more than 85 % of P deposition is brought by combustion aerosols in northern Corsica over the 2008–2011 period. We underline here the need for more deposition measurements in order to better constrain the modeling of such important nutrient sources for the Mediterranean.

Further development of atmospheric and oceanic models should be undertaken in order to account for the mixing and chemical processing of the different aerosol sources in the atmosphere and their effect on nutrient solubility in seawater, and for possible deviations from Redfield ratios in the marine biological compartments. Moreover, oceanic simulations taking into account daily atmospheric deposition of nutrients from dust and combustion over larger time periods would be necessary to assess the variability of the impacts of these sources on marine biogeochemistry.

*Acknowledgements.* The PhD grant of C.R. is funded by CEA. R.W. was funded by a Marie Curie IIF project from European Commission (FABIO, grant 628735). This study contributes to MERMEX (Marine Ecoystem Response Mediterranean Experiment; https://mermex.mio.univ-amu.fr) and ChArMEx (the Chemistry-aerosol Mediterranean Experiment; http:/charmex.lsce.ipsl.fr) projects of the programme MISTRALS (Mediterranean Integrated Studies at Regional and Local Scales; https://www.mistrals-home.org/).

600

**Appendix A: Statistical evaluation of PISCES vertical profiles**

The following Tables and Figures provide an evaluation of the model performances against measurements in different Mediterranean regions obtained from Manca et al. (2004). We provide observed and modeled annually averaged vertical profiles over the year 2005 of phosphate and nitrate and statistical indicators (annual vertical mean and standard deviation, RMSE, normalized and relative bias, Pearson's correlation coefficient R and associated p-value). We point out that the different indicators provide different information. For instance, bias evaluates how mathematically close are modeled and measured values, whereas Pearson's R indicates whether the model reproduces the evolution of $PO_4$ concentration with depth.

These figures show that the model reproduces on average the vertical distribution of phosphate. In some regions such as the Algerian basin, surface concentration is closer to measurements than deep concentrations (Figure A1). In the South Adriatic and in the Gulf of Lions, phosphate concentration below 200 m is close to the measurements (Figures A4 and A5 show normalized bias of -0.09 and 0.0 respectively). On the other hand, very low bias in the South Adriatic region is paired with low Pearson's R (0.30 in the deep layer) whereas $PO_4$ concentration evolution with depth is well reproduced in the Algerian basin (R=0.89 and R=0.97 in the intermediate and deep layers) in spite of a mismatch between measured and modeled values.

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

| Basin | Model mean ($\sigma$) | Data mean ($\sigma$) | RMSE | normalized bias | % bias |
|---|---|---|---|---|---|
| Whole Med. | 0.137 (0.04) | 0.140 (0.1) | 0.06 | -0.01 | -0.4 |
| West | 0.175 (0.03) | 0.215 (0.1) | 0.08 | -0.10 | -23 |
| Adriatic | 0.155 (0.007) | 0.205 (0.04) | 0.06 | -0.13 | -31 |
| Aegean | 0.141 (0.02) | 0.145 (0.1) | 0.10 | -0.01 | -3 |
| Ionian | 0.108 (0.03) | 0.094 (0.04) | 0.03 | 0.07 | 13 |
| Levantine | 0.108 (0.2) | 0.07 (0.02) | 0.04 | 0.23 | 38 |

**Table 1.** Average chlorophyll *a* concentration (spatial standard deviation in brackets) and statistical indicators (spatial RMSE, normalized and relative bias) for different Mediterranean sub–basins (see Figure 5 for the sub–basins limits). Values are calculated from Figure 1. Coastal areas are filtered out as in Figure 1.

| Station | ADIOS | LMDz–INCA (ADIOS period) | LMDz–INCA (2005) |
|---|---|---|---|
| Cap Spartel, Morocco | 6.8 (2.7) | 2.7 (1.5) | 6.3 (4.4) |
| Cap Béar, France | 11 (3.1) | 3.4 (4.8) | 2.1 (3.8) |
| Corsica, France | 28 (4.6) | 3.6 (4.4) | 3.1 (3.9) |
| Mahdia, Tunisia | 24 (2.8) | 3.7 (1.8) | 11.6 (3.3) |
| Lesbos, Greece | 6.0 (2.3) | 3.7 (4.1) | 18.8 (5.2) |
| Crete, Greece | 9.0 (3.2) | 3.3 (2.3) | 8.9 (4.1) |
| Akkuyu, Turkey | 10 (3.2) | 3.7 (4.0) | 14.1 (4.9) |
| Cavo Greco, Cyprus | 4.1 (1.8) | 3.6 (3.1) | 8.6 (4.3) |
| Alexandria, Egypt | 21 (3.3) | 3.4 (2.5) | 8.2 (4.1) |

**Table 2.** Dust deposition fluxes ($g\,m^{-2}\,yr^{-1}$) measured during the ADIOS campaign (derived from Al measured deposition fluxes considering that dust contains 7 % of Al), simulated by the LMDz–INCA model on the ADIOS period (June 2001 - May 2002) and the simulation period (2005). Values in brackets indicate the geometric standard deviations of monthly fluxes (same restrictions on the number of values as in Figure 4.

| Basin | Total P | Pdust | Pcomb | Ref. |
|---|---|---|---|---|
| East | 28 | | | Krom *et al.*, 2010 |
| Whole Med. | 11 (9-21) | 3.6 (1-10) | 7.5 (5-11) | This work |
| West | 9 (6-15) | 1.7 (0-5) | 7.3 (5-11) | This work |
| Adriatic | 6 (4-16) | 0.97 (0-5) | 5.1 | This work |
| Aegean | 11 | 2.0 (0-5) | 9.0 (6-11) | This work |
| Ionian | 40 (27-71) | 20 (5-60) | 20 (10-33) | This work |
| Levantine | 11 (7-18) | 4.3 (1-14) | 7 (4-10) | This work |

**Table 3.** Relative atmospheric contribution (%) to total $PO_4$ supply in different sub–basins (atmospheric inputs/(atmospheric inputs + riverine inputs + Gibraltar inputs)) according to the model. The values in parentheses show the minimum and maximum monthly contributions over the year when variability is more than 3 %. The sub basins are described in Figure 5. Values from Krom et al. (2010) also include river inputs.

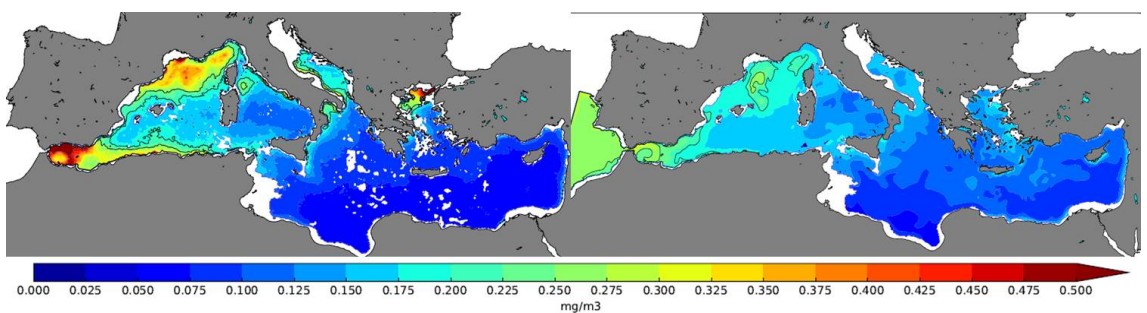

**Figure 1.** Satellite map of average surface chlorophyll *a* concentration from Bosc et al. (2004) (1997–2004, left) and modeled average surface chlorophyll *a* concentration (right). Model and satellite data are filtered for coastal waters (white areas). Additional white areas on the satellite maps are lack of data.

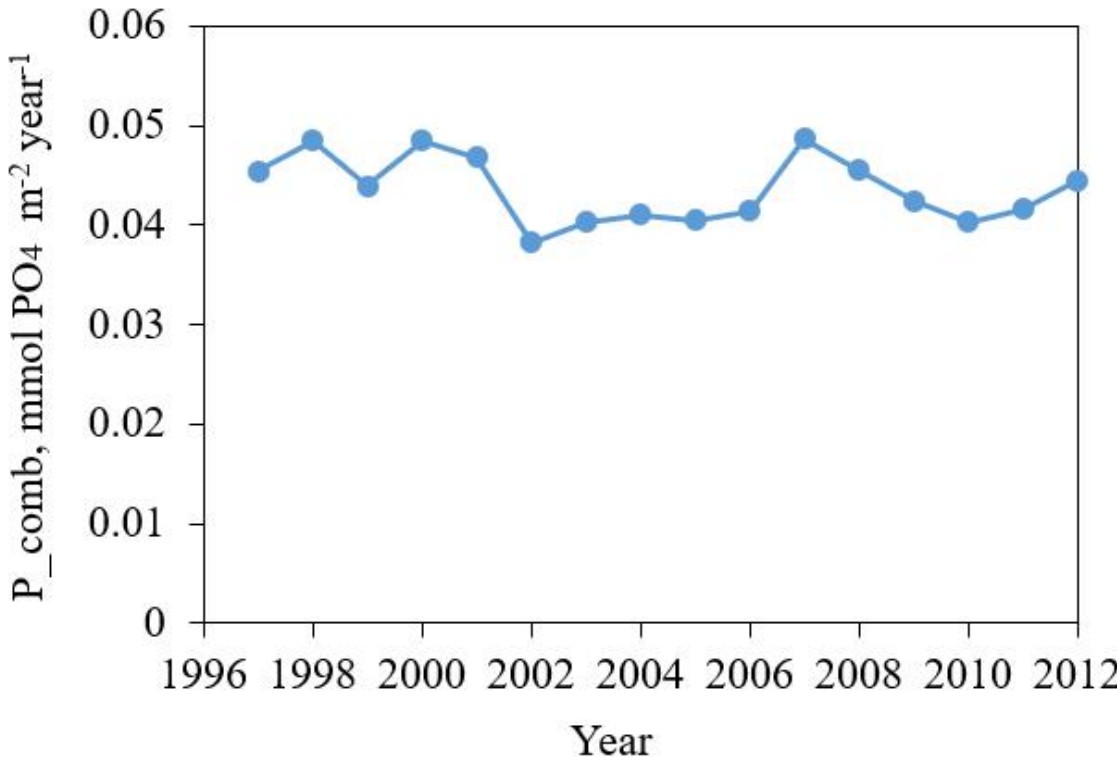

**Figure 2.** Plot of yearly average *Pcomb* deposition over the Mediterranean for the 1997–2012 period.

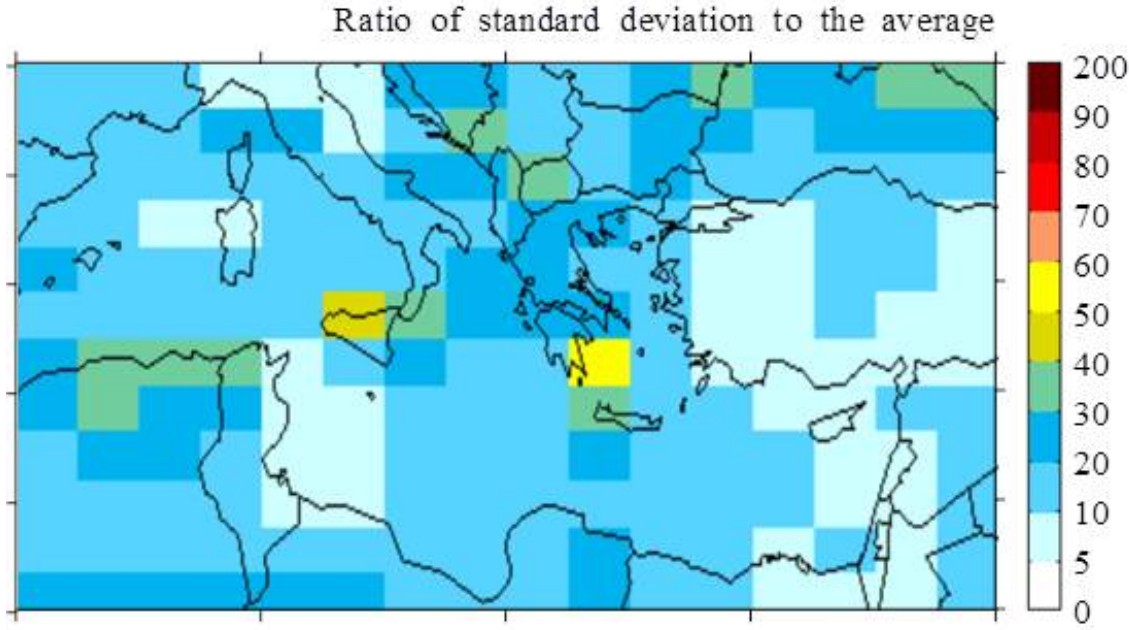

**Figure 3.** Map of the standard deviation to average ratio of *Pcomb* deposition over the Mediterranean for the 1997–2012 period.

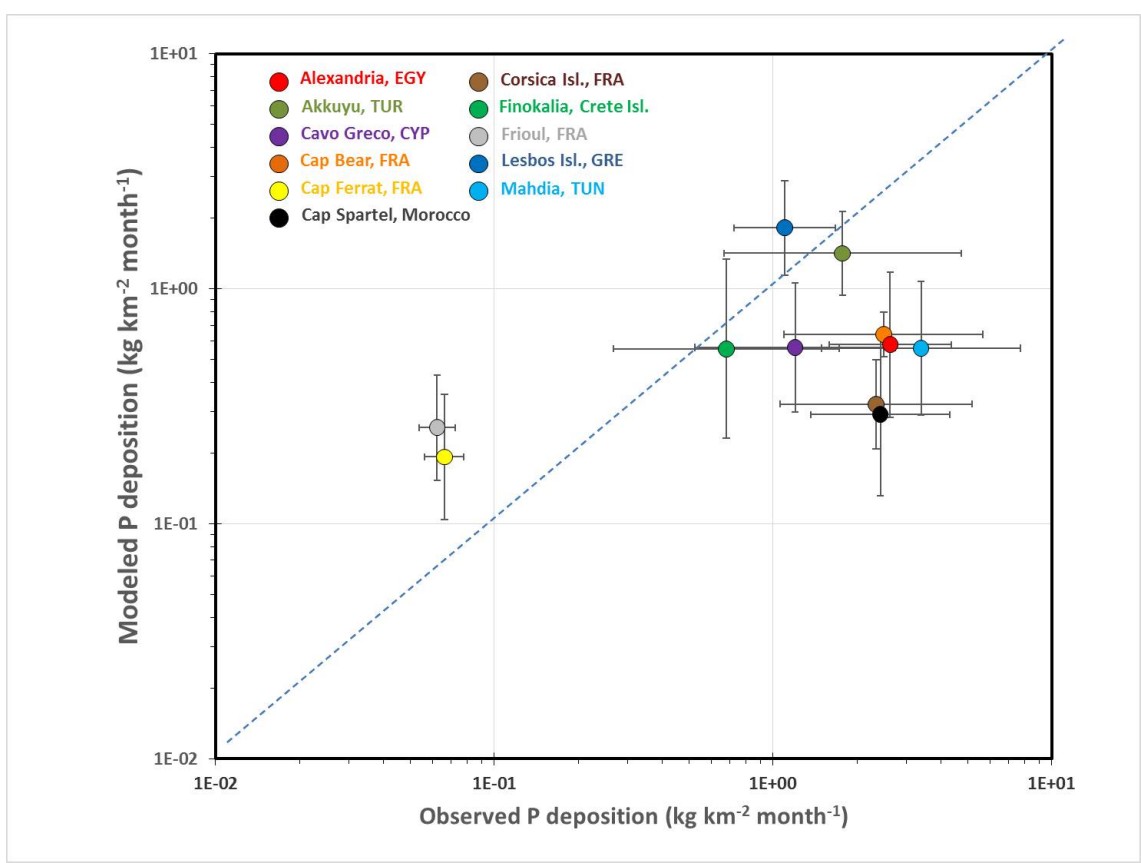

**Figure 4.** Comparison of modeled and observed monthly geometric mean of total P (*Pdust + Pcomb*) deposition fluxes at the 9 ADIOS stations (Guieu et al., 2010) and soluble $PO_4$ at Frioul and Cap Ferrat stations (de Fommervault et al., 2015). Each point is the geometric mean of monthly observed and modeled values at the given station over 1 year, namely 2005 for the model and June 2001–May 2002 for the ADIOS observations (only 6 values are available at Alexandria to compute the observed mean and standard deviation, 10 at Mahdia, and 11 at Finokalia) and between 2007 and 2012 for the observations at Frioul and Cap Ferrat stations. Error bars represent the geometric standard deviation on model (y—axis) and measurements (x—axis).

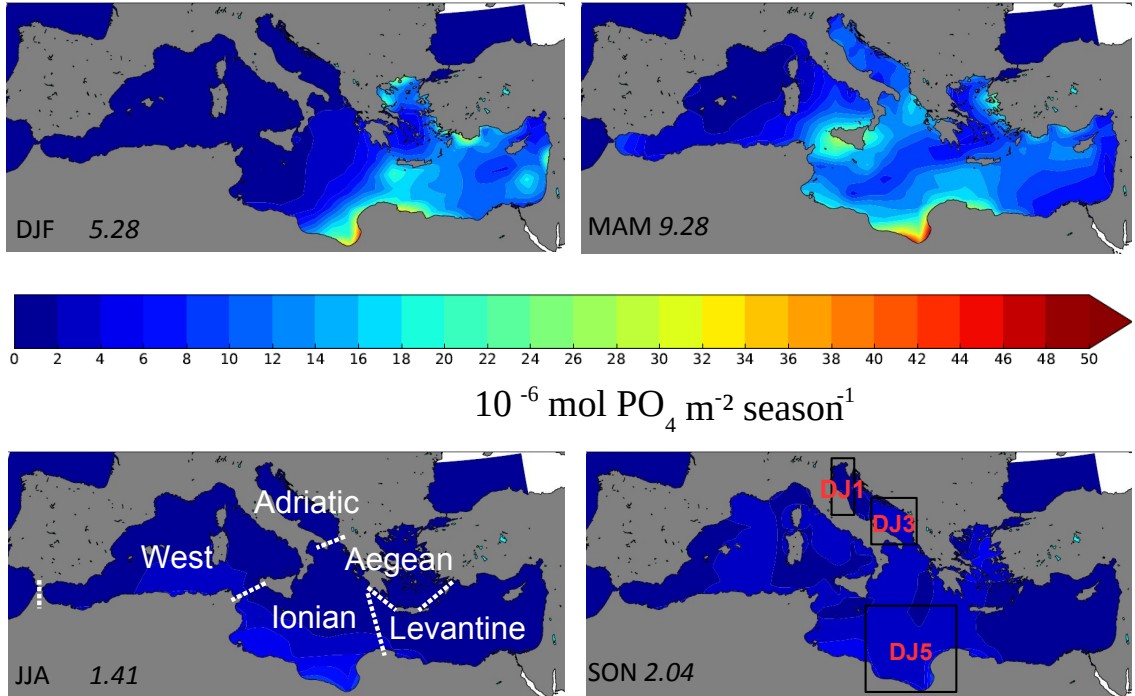

**Figure 5.** Total seasonal desert dust derived soluble phosphorus deposition (*Pdust*, in $10^{-6}$ molPO$_4$ m$^{-2}$ season$^{-1}$) over each season of the year 2005 (molar flux is calculated as mass flux/phosphorus molar weight) from the LMDz–INCA model. Numbers on the maps are the average seasonal deposition fluxes over the whole basin in $10^{-6}$ molPO$_4$ m$^{-2}$ season$^{-1}$. In the Summer (JJA) deposition map, we display the different sub regions referred to in the text. In the Automn (SON) map, we display sub regions as defined in Manca et al. (2004): DJ1 is the North Adriatic region, DJ3 is the South Adriatic region and DJ5 is the South Ionian region.

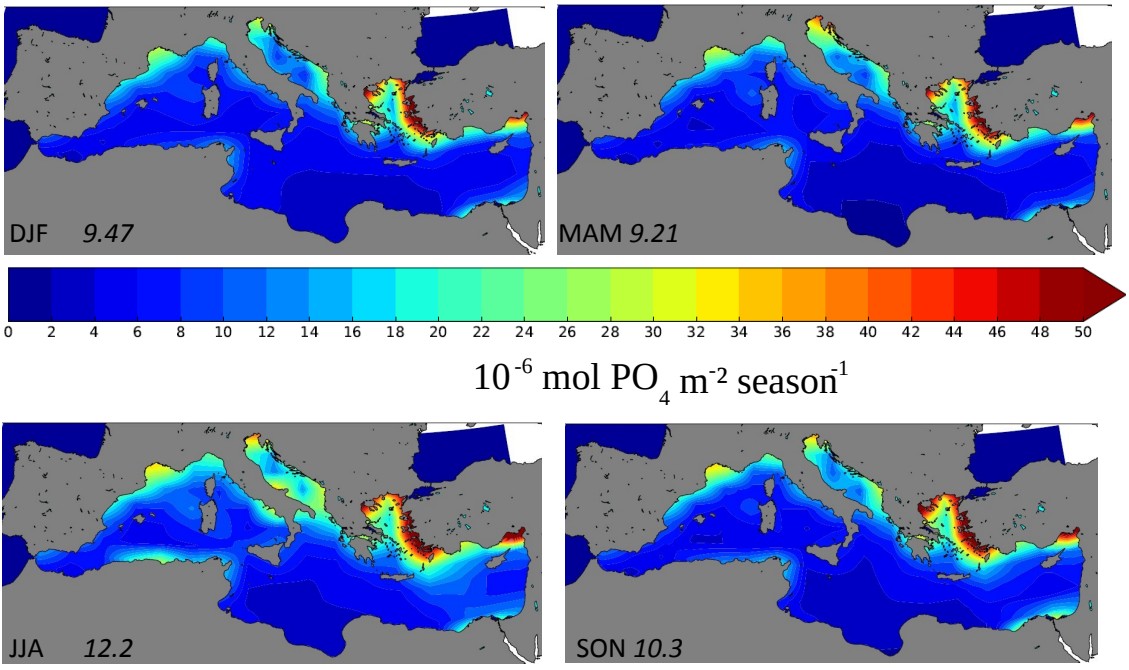

**Figure 6.** Total seasonal combustion–derived soluble phosphorus deposition (*Pcomb* in $10^{-6}$ $molPO_4$ $m^{-2}$ $season^{-1}$) over each season of the year 2005 (molar flux is calculated as mass flux/phosphorus molar weight) from the LMDz–INCA model. Numbers on the maps are the average seasonal deposition fluxes over the basin in $10^{-6}$ $molPO_4$ $m^{-2}$ $season^{-1}$.

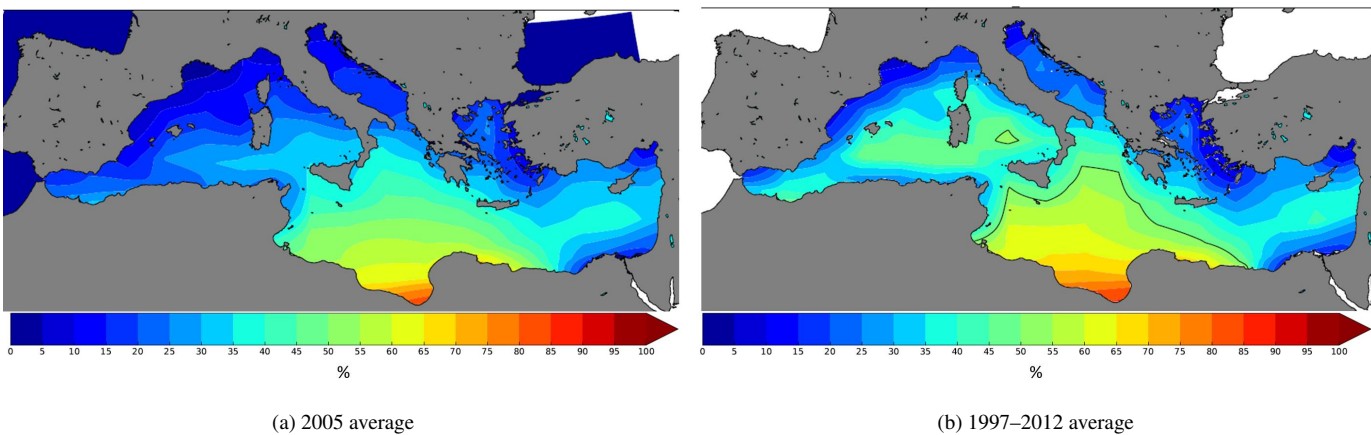

(a) 2005 average

(b) 1997–2012 average

**Figure 7.** Map of average *Pdust* proportion in total P deposition for 2005 (left) and 1997–2012 (right). The black line on the right map represents the 50 % *Pdust* proportion limit on the average of 1997–2012.

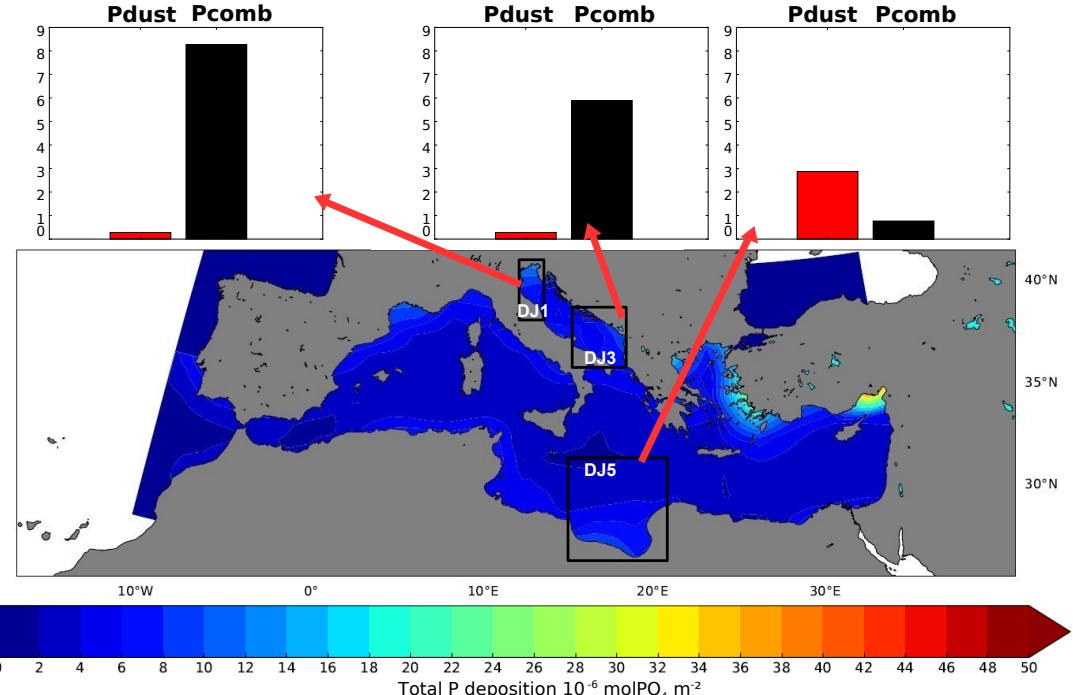

**Figure 8.** Map of total $PO_4$ deposition from both *Pdust* and *Pcomb* ($10^{-6}$ $molPO_4$ $m^{-2}$) for June 2005. Red and black bars represent average $PO_4$ deposition (in $10^{-6}$ $molPO_4$ $m^{-2}$) from the two sources in each framed area. The limits of the areas are described in Manca et al. (2004) and Figure 5. There is no atmospheric deposition modeled in the Marmara and Black Seas.

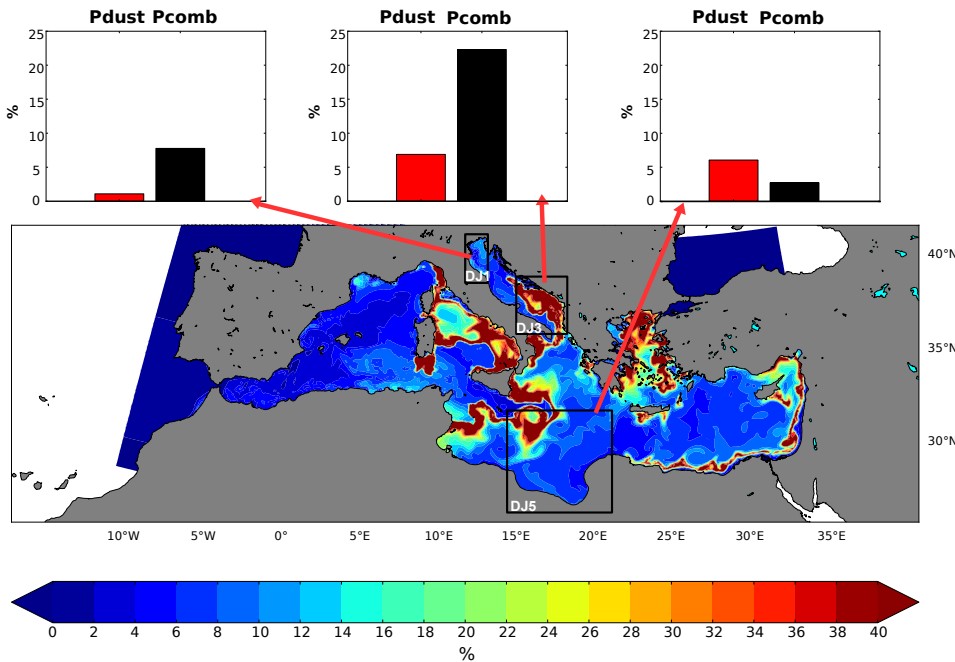

**Figure 9.** Map of maximal relative effects of total (*Pdust*+*Pcomb*) deposition in June 2005 (on a daily basis) on the surface phosphate concentration (0–10 m). The reference PO$_4$ concentration values are taken from the REF simulation without atmospheric phosphate deposition. Red and black bars represent average relative effects (%) within the framed areas for each P source. The limits of the areas are described in Manca et al. (2004) and Figure 5. There is no atmospheric deposition modeled in the Marmara and Black Seas.

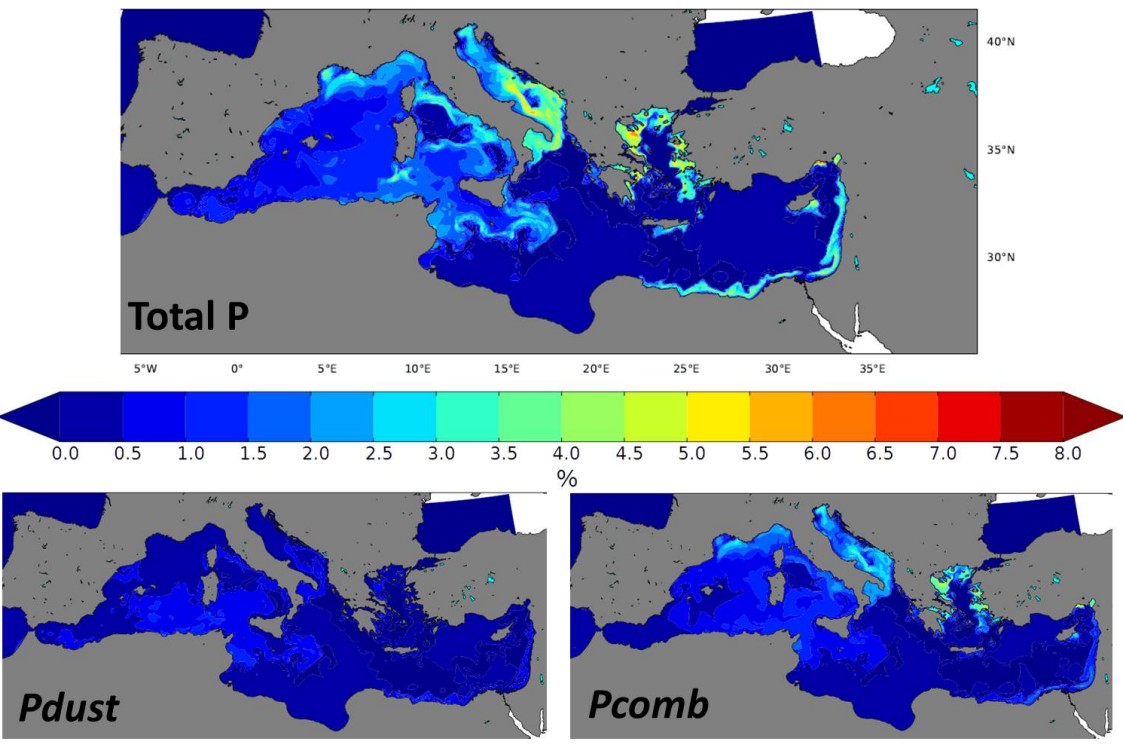

**Figure 10.** Average relative effects of total P, *Pdust* and *Pcomb* deposition on surface (0–10 m) chlorophyll *a* concentration for June 2005. There is no atmospheric deposition modeled in the Marmara and Black Seas.

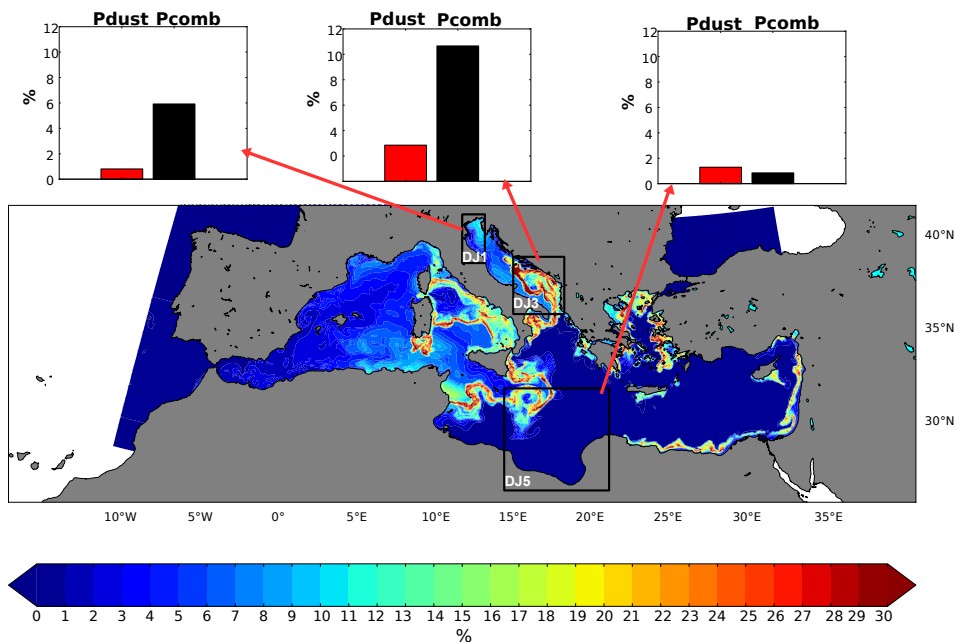

**Figure 11.** Map of maximal relative effects of total (*Pdust*+*Pcomb*) deposition on primary production in the surface Mediterranean (0–10 m) in June 2005 (on a daily basis). The reference primary production concentration values are taken from the REF simulation without atmospheric phosphate deposition. Barplots represent average relative effects of each source (%) within the framed areas excluding land. The limits of the areas are described in Manca et al. (2004). There is no atmospheric deposition modeled in the Marmara and Black Seas.

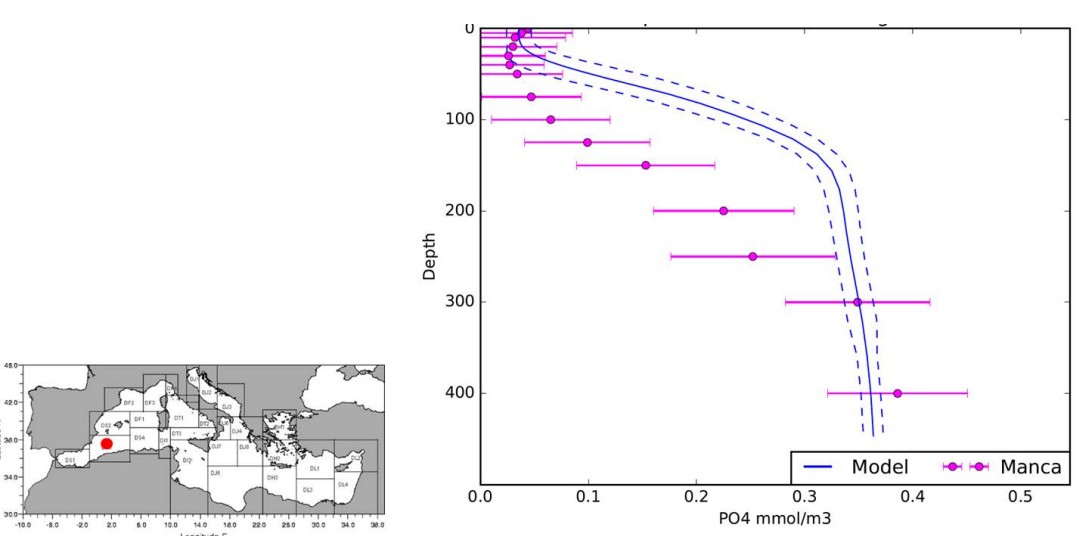

| Depth | Model mean | Data mean | Model std | Data std | RMSE | normalized bias | % bias | Pearson's R | P-value |
|-------|-----------|-----------|-----------|----------|------|-----------------|--------|-------------|---------|
| **0-50** | 0.045 | 0.033 | 0.014 | 0.006 | 0.02 | 0.16 | 27.02 | -0.64 | 0.17 |
| **50-200** | 0.23 | 0.080 | 0.079 | 0.043 | 0.16 | 0.48 | 65.27 | 0.89 | 0.04 |
| **200-500** | 0.35 | 0.30 | 0.009 | 0.067 | 0.07 | 0.07 | 12.78 | 0.97 | 0.03 |

**Figure A1.** Annually averaged $PO_4$ vertical profile in the Algerian sub–basin (see map and Manca et al. (2004)) and statistical indicators over different depths. Model values are the blue line (2005 average, Ntot simulation from Richon et al., 2017) and measured values from Manca et al. (2004) are in pink. Horizontal bars and dashed lines indicate spatial standard deviation of observations and model results respectively.

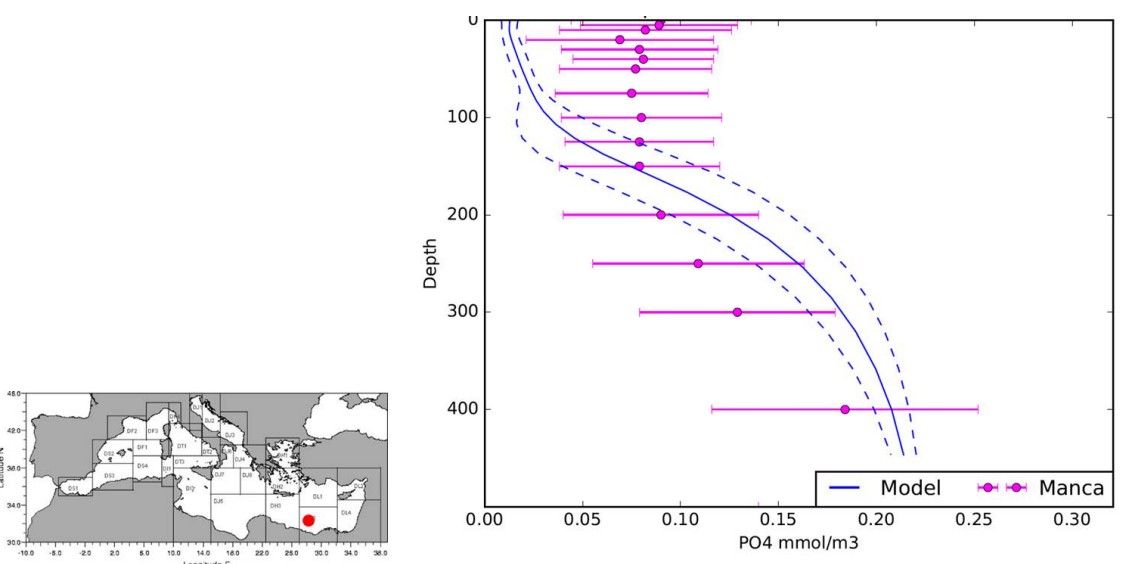

| Depth | Model mean | Data mean | Model std | Data std | RMSE | normalized bias | % bias | Pearson's R | P-value |
|-------|-----------|-----------|-----------|----------|------|-----------------|--------|-------------|---------|
| 0-50 | 0.014 | 0.082 | 0.0017 | 0.007 | 0.07 | -0.71 | -483.57 | -0.31 | 0.55 |
| 50-200 | 0.040 | 0.078 | 0.020 | 0.002 | 0.04 | -0.32 | -95.09 | 0.55 | 0.34 |
| 200-500 | 0.17 | 0.13 | 0.030 | 0.035 | 0.04 | 0.14 | 24.20 | 0.94 | 0.06 |

**Figure A2.** Annually averaged $PO_4$ vertical profile in the South Levantine sub–basin (see map and Manca et al. (2004)) and statistical indicators over different depths. Model values are the blue line (2005 average, Ntot simulation from Richon et al., 2017) and measured values from Manca et al. (2004) are in pink. Horizontal bars and dashed lines indicate spatial standard deviation of observations and model results respectively.

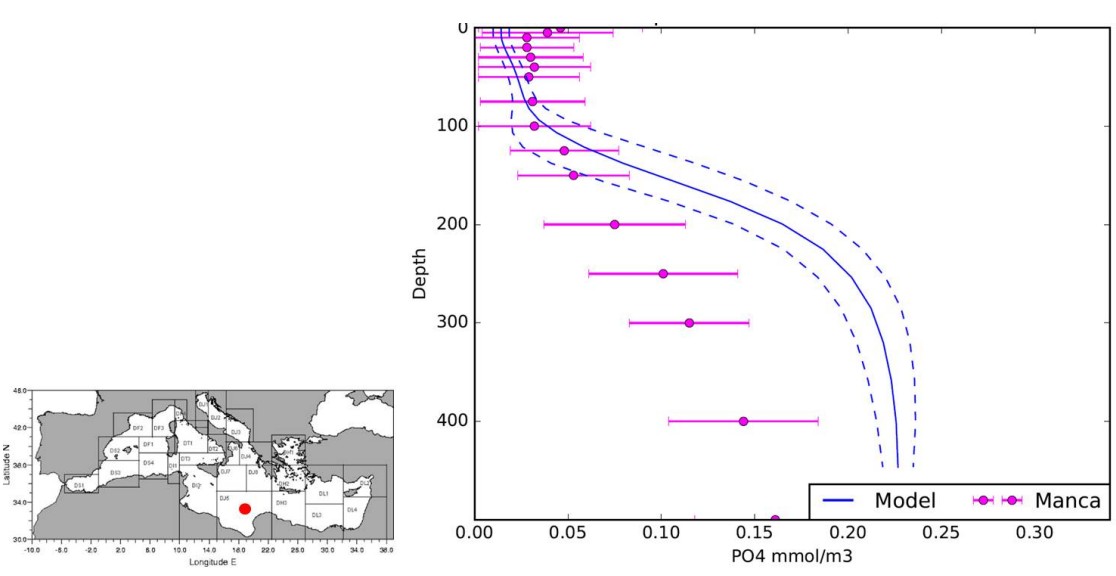

| Depth | Model mean | Data mean | Model std | Data std | RMSE | normalized bias | % bias | Pearson's R | P-value |
|---|---|---|---|---|---|---|---|---|---|
| **0-50** | 0.016 | 0.034 | 0.003 | 0.007 | 0.02 | -0.35 | -107.36 | -0.40 | 0.43 |
| **50-200** | 0.050 | 0.039 | 0.028 | 0.010 | 0.02 | 0.13 | 22.86 | 0.96 | 0.01 |
| **200-500** | 0.20 | 0.11 | 0.023 | 0.025 | 0.09 | 0.30 | 46.03 | 0.96 | 0.04 |

**Figure A3.** Annually averaged PO$_4$ vertical profile in the South Ionian sub–basin (see map and Manca et al. (2004)) and statistical indicators over different depths. Model values are the blue line (2005 average, Ntot simulation from Richon et al., 2017) and measured values from Manca et al. (2004) are in pink. Horizontal bars and dashed lines indicate spatial standard deviation of observations and model results respectively.

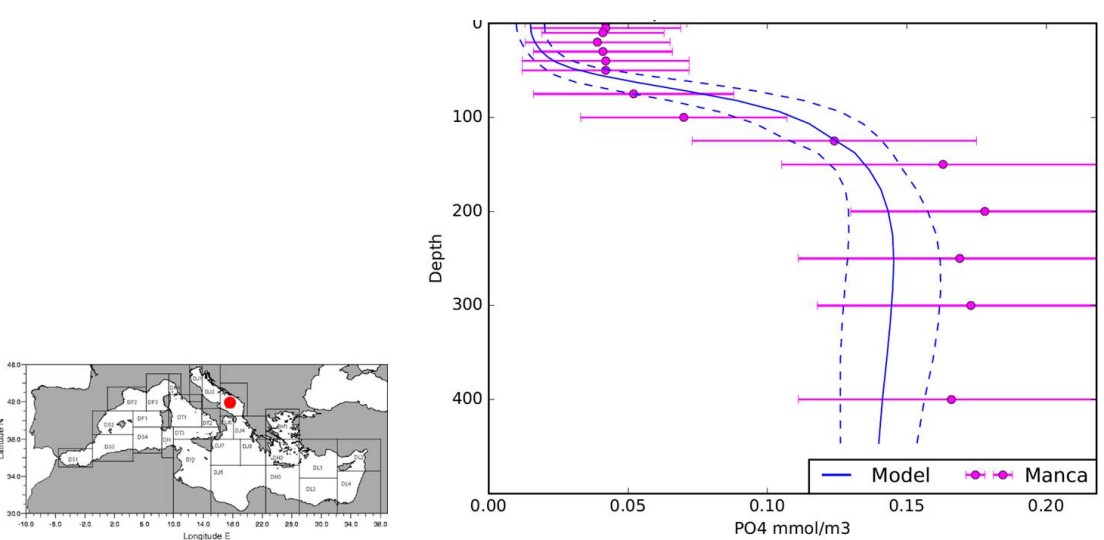

| Depth | Model mean | Data mean | Model std | Data std | RMSE | normalized bias | % bias | Pearson's R | P-value |
|---|---|---|---|---|---|---|---|---|---|
| **0-50** | 0.018 | 0.041 | 0.003 | 0.001 | 0.02 | -0.40 | -135.84 | 0.15 | 0.78 |
| **50-200** | 0.096 | 0.090 | 0.037 | 0.046 | 0.02 | 0.03 | 5.66 | 0.86 | 0.06 |
| **200-500** | 0.14 | 0.17 | 0.002 | 0.005 | 0.03 | -0.09 | -19.43 | 0.30 | 0.70 |

**Figure A4.** Annually averaged PO$_4$ vertical profile in the South Adriatic sub–basin (see map and Manca et al. (2004)) and statistical indicators over different depth. Model values are the blue line (2005 average, Ntot simulation from Richon et al., 2017) and measured values from Manca et al. (2004) are in pink. Horizontal bars and dashed lines indicate spatial standard deviation of observations and model results respectively.

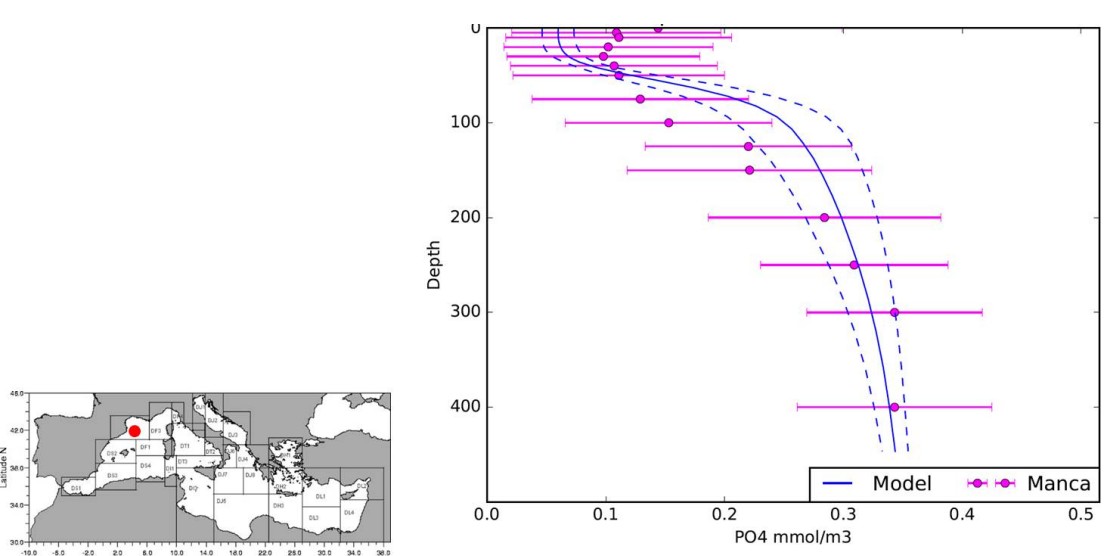

| Depth | Model mean | Data mean | Model std | Data std | RMSE | normalized bias | % bias | Pearson's R | P-value |
|---|---|---|---|---|---|---|---|---|---|
| **0-50** | 0.066 | 0.11 | 0.010 | 0.015 | 0.05 | -0.26 | -68.59 | -0.31 | 0.56 |
| **50-200** | 0.23 | 0.17 | 0.057 | 0.046 | 0.07 | 0.15 | 26.15 | 0.86 | 0.06 |
| **200-500** | 0.32 | 0.32 | 0.015 | 0.025 | 0.01 | 0.00 | -0.48 | 0.93 | 0.07 |

**Figure A5.** Annually averaged PO$_4$ vertical profile in the Gulf of Lions sub–basin (see map and Manca et al. (2004)) and statistical indicators over different depth. Model values are the blue line (2005 average, Ntot simulation from Richon et al., 2017) and measured values from Manca et al. (2004) are in pink. Horizontal bars and dashed lines indicate spatial standard deviation of observations and model results respectively.

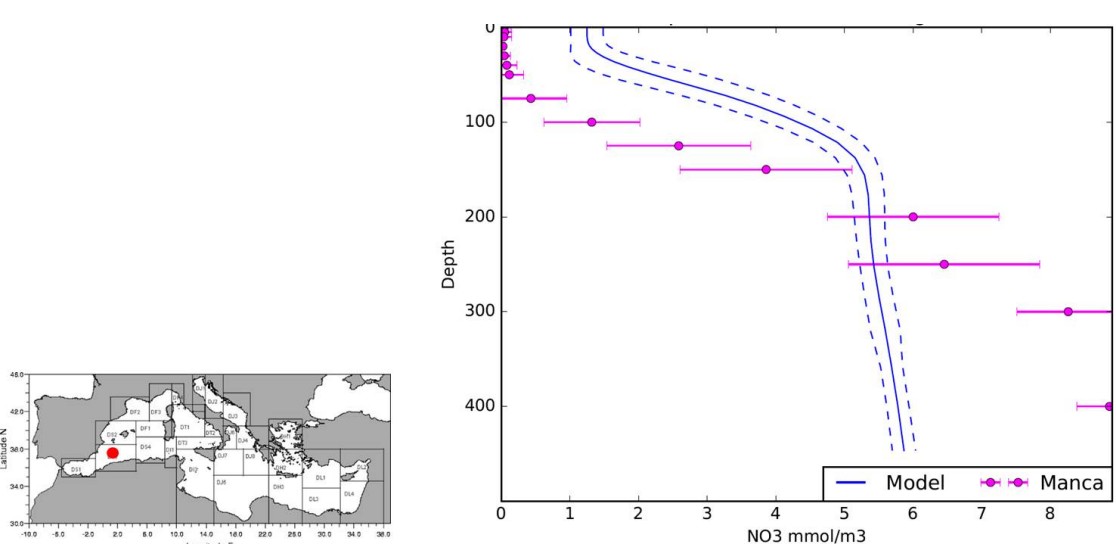

| Depth | Model mean | Data mean | Model std | Data std | RMSE | normalized bias | % bias | Pearson's R | P-value |
|---|---|---|---|---|---|---|---|---|---|
| **0-50** | 1.38 | 0.05 | 0.20 | 0.02 | 1.35 | 0.94 | 97 | 0.83 | 0.04 |
| **50-200** | 4.02 | 1.66 | 1.11 | 1.39 | 2.44 | 0.42 | 59 | 0.91 | 0.03 |
| **200-500** | 5.53 | 7.39 | 0.16 | 1.20 | 2.14 | -0.14 | -34 | 0.93 | 0.07 |

**Figure A6.** Annually averaged NO$_3$ vertical profile in the Algerian sub–basin (see map and Manca et al. (2004)) and statistical indicators over different depths. Model values are the blue line (2005 average, Ntot simulation from Richon et al., 2017) and measured values from Manca et al. (2004) are in pink. Horizontal bars and dashed lines indicate spatial standard deviation of observations and model results respectively.

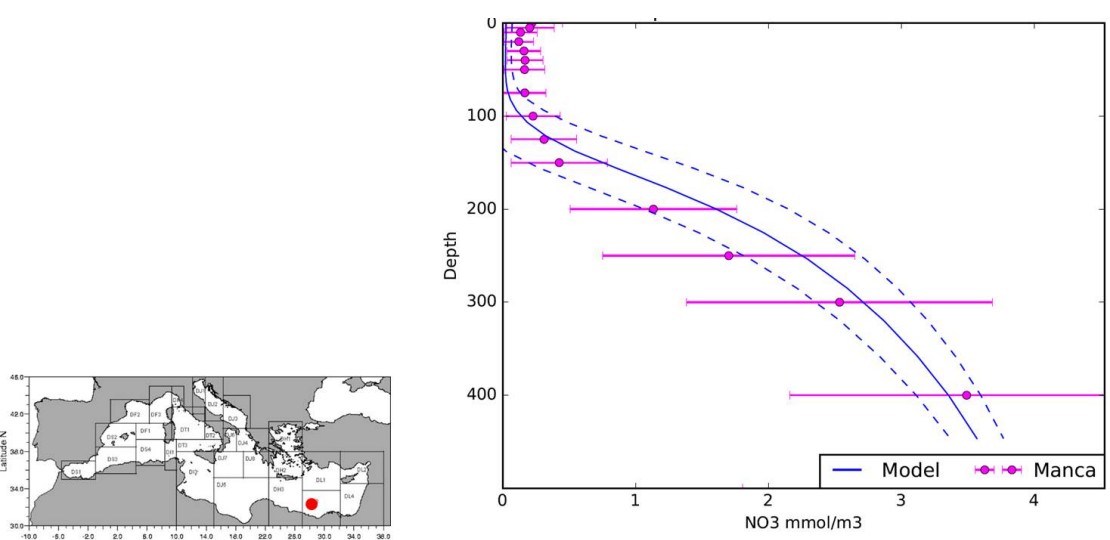

| Depth | Model mean | Data mean | Model std | Data std | RMSE | normalized bias | % bias | Pearson's R | P-value |
|-------|-----------|-----------|-----------|----------|------|-----------------|--------|-------------|---------|
| **0-50** | 0.02 | 0.17 | 0.00 | 0.03 | 0.15 | -0.75 | -606 | 0.49 | 0.33 |
| **50-200** | 0.27 | 0.26 | 0.27 | 0.10 | 0.18 | 0.01 | 3 | 0.99 | 0.00 |
| **200-500** | 2.48 | 2.22 | 0.64 | 0.89 | 0.38 | 0.06 | 11 | 0.99 | 0.01 |

**Figure A7.** Annually averaged NO$_3$ vertical profile in the South Levantine sub–basin (see map and Manca et al. (2004)) and statistical indicators over different depths. Model values are the blue line (2005 average, Ntot simulation from Richon et al., 2017) and measured values from Manca et al. (2004) are in pink. Horizontal bars and dashed lines indicate spatial standard deviation of observations and model results respectively.

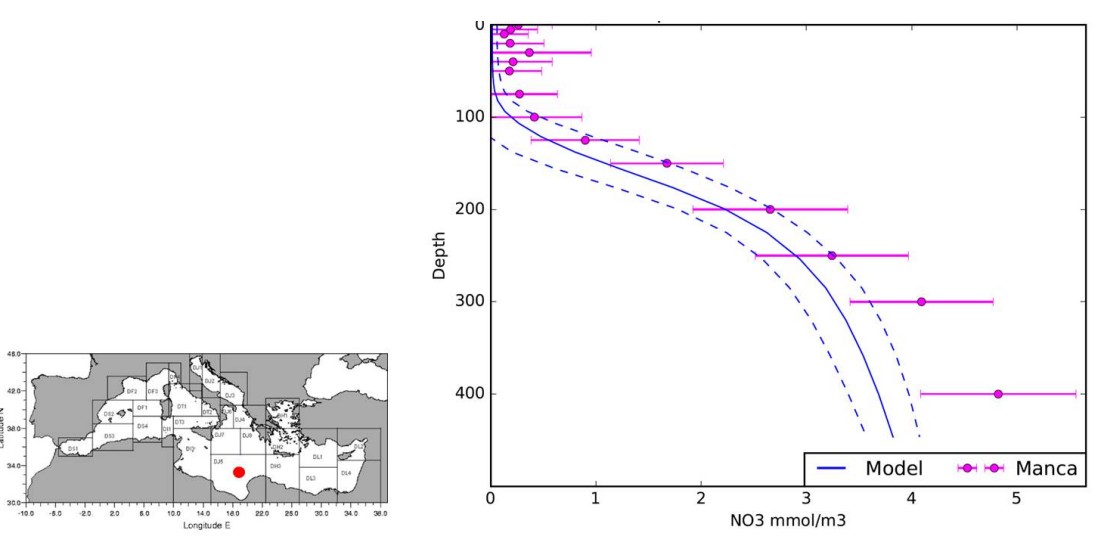

| Depth | Model mean | Data mean | Model std | Data std | RMSE | normalized bias | % bias | Pearson's R | P-value |
|---|---|---|---|---|---|---|---|---|---|
| **0-50** | 0.015 | 0.224 | 0.001 | 0.075 | 0.22 | -0.87 | -1394 | 0.14 | 0.78 |
| **50-200** | 0.381 | 0.687 | 0.402 | 0.552 | 0.34 | -0.29 | -81 | 1.00 | 0.00 |
| **200-500** | 3.020 | 3.704 | 0.534 | 0.824 | 0.76 | -0.10 | -23 | 0.98 | 0.02 |

**Figure A8.** Annually averaged NO$_3$ vertical profile in the South Ionian sub–basin (see map and Manca et al. (2004)) and statistical indicators over different depths. Model values are the blue line (2005 average, Ntot simulation from Richon et al., 2017) and measured values from Manca et al. (2004) are in pink. Horizontal bars and dashed lines indicate spatial standard deviation of observations and model results respectively.

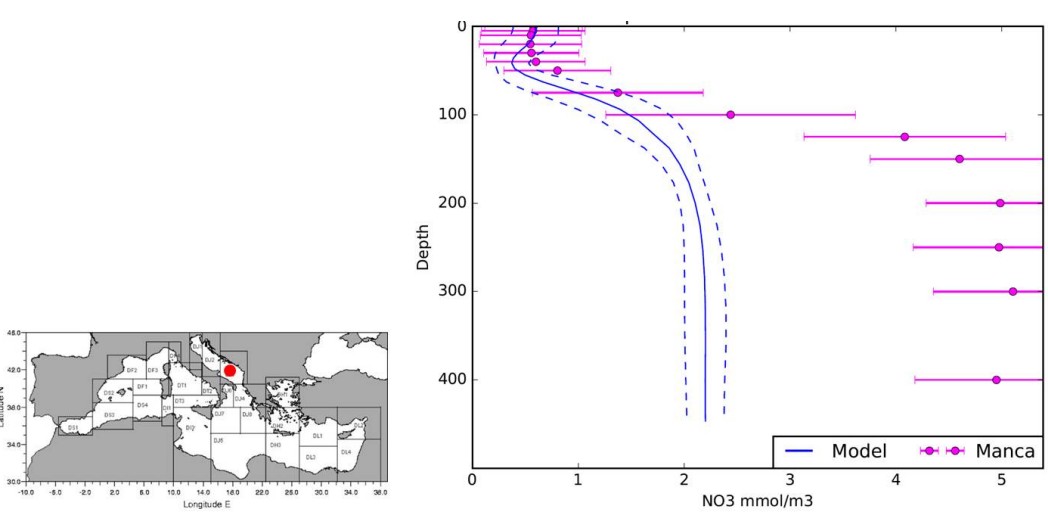

| Depth | Model mean | Data mean | Model std | Data std | RMSE | normalized bias | % bias | Pearson's R | P-value |
|-------|-----------|-----------|-----------|----------|------|-----------------|--------|-------------|---------|
| **0-50** | 0.52 | 0.57 | 0.09 | 0.02 | 0.11 | -0.05 | -10 | -0.40 | 0.43 |
| **50-200** | 1.31 | 2.66 | 0.54 | 1.48 | 1.66 | -0.34 | -103 | 0.95 | 0.01 |
| **200-500** | 2.17 | 5.00 | 0.04 | 0.06 | 2.84 | -0.40 | -131 | 0.23 | 0.77 |

**Figure A9.** Annually averaged NO$_3$ vertical profile in the South Adriatic sub–basin (see map and Manca et al. (2004)) and statistical indicators over different depth. Model values are the blue line (2005 average, Ntot simulation from Richon et al., 2017) and measured values from Manca et al. (2004) are in pink. Horizontal bars and dashed lines indicate spatial standard deviation of observations and model results respectively.

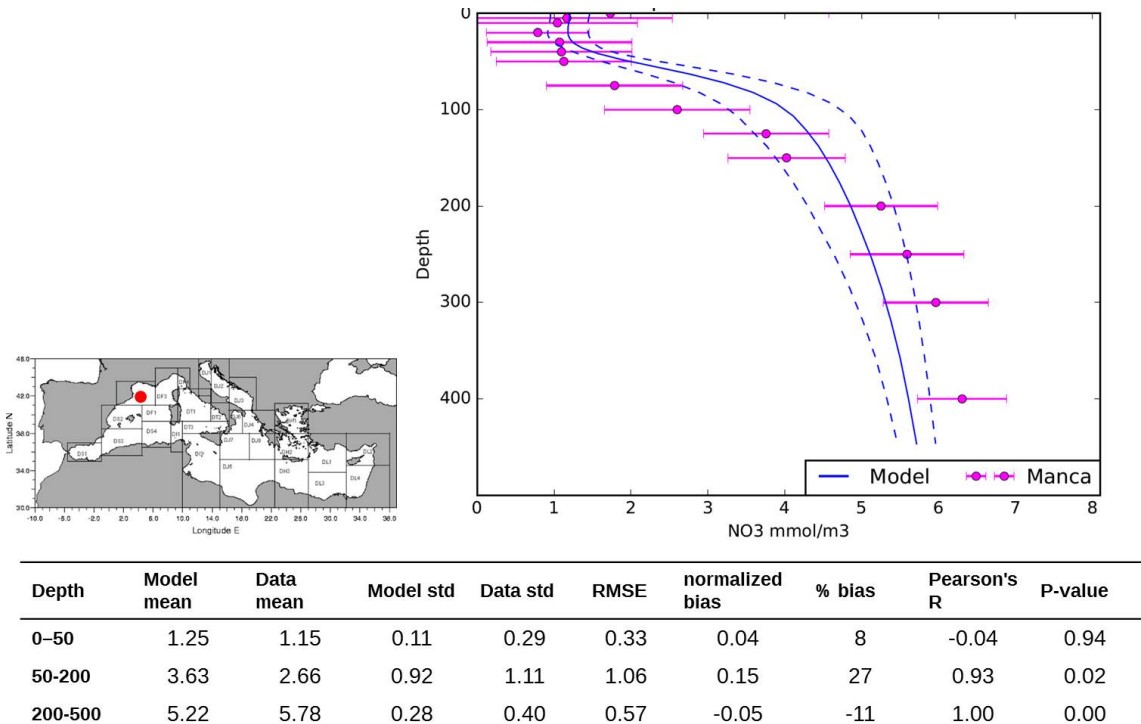

| Depth | Model mean | Data mean | Model std | Data std | RMSE | normalized bias | % bias | Pearson's R | P-value |
|---|---|---|---|---|---|---|---|---|---|
| **0–50** | 1.25 | 1.15 | 0.11 | 0.29 | 0.33 | 0.04 | 8 | -0.04 | 0.94 |
| **50-200** | 3.63 | 2.66 | 0.92 | 1.11 | 1.06 | 0.15 | 27 | 0.93 | 0.02 |
| **200-500** | 5.22 | 5.78 | 0.28 | 0.40 | 0.57 | -0.05 | -11 | 1.00 | 0.00 |

**Figure A10.** Annually averaged NO$_3$ vertical profile in the Gulf of Lions sub–basin (see map and Manca et al. (2004)) and statistical indicators over different depth. Model values are the blue line (2005 average, Ntot simulation from Richon et al., 2017) and measured values from Manca et al. (2004) are in pink. Horizontal bars and dashed lines indicate spatial standard deviation of observations and model results respectively.