# Peer review of "Modeling the biogeochemical impact of atmospheric phosphate deposition from desert dust and combustion sources to the Mediterranean Sea"

_Biogeosciences, 2017_

## Referee Comment (RC1) · Anonymous Referee #1 · 24 Jul 2017

The present manuscript "Modeling the biogeochemical impact of atmospheric phosphate deposition from desert dust and combustion sources to the Mediterranean Sea" proposes an analysis of the impact of phosphorus atmospheric deposition comparing different sources: namely desert dust (Pdust) and combustion sources (Pcomb). The idea is very interesting and useful because the two sources are, in principle, characterized by different spatial and temporal distributions. But, the main weakness of the manuscript is, in my opinion, the insufficient skill of the global atmospheric model LMDz-INCA in reproducing correctly the amount of dust deposition fluxes and

its spatial and temporal variability for the Mediterranean area. Authors cite another model, the higher resolution ALADIN-Climat (Nabat et al. 2012), used for a companion paper (Richon et al prog ocean. 2017), which gives higher deposition rate. I think that it is necessary to add also a test with the ALADIN-Climat model (equipped with the proper phosphorus deposition model), in order to have at least an ensemble composed by two members, this would make results more robust. Moreover, the choice of selecting only the year 2005 given the high variability of Pdust, is not clear to me. This high variability is important and its impact should be estimated. Therefore, I suggest that the present manuscript can be published only after major revisions of the simulation protocol.

**Main comments**
(Text from Authors in quotes, comments indented)

**ABSTRACT**
**lines 18-20:** "The impact of the different sources of phosphate on the biogeochemical cycles is remarkably different and should be accounted for in modeling studies."

   This sentence is, in my opinion, not clear, "remarkably different" with respect
   to what?

The oceanic model
**Pg 4. line113:line 115** : "The model satisfyingly reproduces the vertical distribution of nutrients in the basin and the main productive zones that are the Alboran Sea, the Gulf of Lions and most coastal areas (see appendix)."

   The comparison/validation shown in the appendix appears quite subjective,
   no objective statistical indicators are provided.

**Results section**
**Evaluation of P deposition fluxes.**
**Pg6, line 198: Pg 7, line 200**: "We were able to compare the dust deposition flux modeled with LMDz–INCA used to derive Pdust deposition over the ADIOS sampling period with the measurements. The comparison is shown in Table 1. The dust fluxes produced by the model are realistic."

I plotted results reported in Tab 1. See figure attached (x-axis stations, y-axis dust deposition, units are g m-2yr-1). In my opinion dust fluxes produced by the model (brown line; MODEL "ADIOS period") compared to data (blue line; DATA "ADIOS period") are very different. In particular there is a strong underestimation of the model, about an order of magnitude, and the spatial variability across stations is absent in the model. So the sentence "The dust fluxes produced by the model are realistic", should be substituted by something like "model presents a strong underestimation compared to data and it is not able to represent the spatial variability of the data". Clearly, as stated by Authors, the dataset available is not enough, and continuous times series at different stations should be used to corroborate the model. But, given this situation, the usage of another atmospheric model, for example the ALADIN-Climat is, in my opinion, mandatory. A higher resolution model would allow for more robust results in terms of spatial gradients of dust deposition also.

**Characterization of phosphate deposition from the different sources**

**Pg 7, lines 221: lines 226**: also in this case the estimates for the total deposition flux by the model seem low. In a recent paper [Powley et al. (2017), Global Biogeochem. Cycles, 31, 1010-1031] Authors report atmospheric deposition rates of 0.16 109 mol/yr WMS and of 0.38 109 mol/yr

EMS (see their Tab. 3). In the present manuscript the estimates are much lower. Given the lack of data it is difficult to r each a conclusion, but anyway this discrepancy raises the question of how robust is the discussion on spatial gradients if the average values present such an uncertainty.

**Pg 8, line 239: line 226** : "However, riverine inputs are the dominant external source of phosphate for almost all Mediterranean regions"

Given the uncertainty on phosphorus deposition, and apparently its under-estimation, this sentence appears not demonstrated.

**Discussion**

**Pg 11, line 363: line 365** : "The atmospheric model LMDz–INCA has a low resolution given the regional Mediterranean scale: Pdust deposition forcing has 280x193 grid points globally and âĹij500 grid points covering the Mediterranean, and Pcomb forcing has 144x143 grid points in total and âĹij200 grid points covering the Mediterranean. These forcings reproduce well the average deposition patterns at the basin scale but may not be reliable when analyzing small scale deposition patterns."

The statement that the global forcing reproduces well the average deposition should be somehow proved.

**Pg 12, line 381:line 384** : "Natural dust emissions, transport and deposition to the Mediterranean are shown to be highly variable from a year to the next (e.g. Moulin et al., 1997; Laurent et al., 2008; Vincent et al., 2016) so that the relative contributions of Pcomb and Pdust may also vary."

Authors focused on an average (or median) year, namely the 2005. But given the high inter-annaul variability, at least of Pdust, what is the meaning

of such a choice? It would be better to consider many years and analyse the temporal variability to have a better quantification of the reality? What is the role of extremes?

[Figure]

**Fig. 1.** Dust Deposition fluxes from Tab1, present manuscript.

---

## Short Comment (SC1) · 25 Jul 2017

Referee#1 is right to consider that having another set of forcings with higher resolution would represent an undeniable improvement for our modelling efforts. Unfortunately this simulation is not conceivable at short term, because such forcings are not available yet. For instance, the Aladin-Climat model used in our previous study, presently does not simulate Phosphate from combustion (it has only Pdust at the moment), and this product will not be available soon since this development requires time.

[Figure]

In order to make progress anyway on our scientific research, we were forced, for a preliminary study to use low resolution forcings, a classical strategy. We comment the limits of this approach in the manuscript and encourage for revisiting our conclusions with more refined forcings in the future. However we consider that our study has revealed some new interesting results, such the spatial difference for the impact of PCOMB and Pdust and their relative quantification, that represent new informations that deserve to be presented to our scientific community, as they have an importance for the modelling and the functioning of the biogeochemical cycle of the Mediterranean sea. We hope as well that this preliminary study will motivate atmospheric regional modelling group for producing high resolution more appropriate aerosols deposition field soon.

---

## Referee Comment (RC2) · Anonymous Referee #2 · 8 Sep 2017

Anonymous Review for "Modeling the biogeochemical impact of atmospheric phosphate deposition from desert dust and combustion sources to the Mediterranean Sea"

General comments

In this work, the authors assess how modeled phosphate deposition output from dust and combustion aerosols can affect the phosphate fluxes into the surface waters of the Mediterranean Sea. The oligotrophic Mediterranean is phosphorus stressed, limited, or co-limited in certain regions/species, and atmospheric deposition may be an important

source of this nutrient. Given high anthropogenic impact on aerosols in this region, and potential future enhancements in surface water stratification, this is a topic worthy of study.

The methodology in this paper was good in most cases, and some of the important uncertainties were discussed very thoroughly. I have pointed out in the specific comments several places where the manuscript requires further explanation of the methodology. My main issue is that, in my opinion, the importance of this study was overstated, and that a few key uncertainties in the findings were downplayed too much (e.g., nutrient co-limitation, the influence of soluble organic P in deposition, non-Redfieldian marine biogeochemical dynamics, and some important model uncertainties).

Because of this latter concern, I suggest the authors proceed in one of two ways: 1) Scale back the conclusions substantially, to focus on the differences between model-estimated Pcomb and Pdust deposition and their potential implications in a (more clearly-emphasized) highly-simplified Redfieldian ocean, or 2) Maintain the scope that the authors do now, but also present results from non-Redfield experiments with prognostic biogeochemistry (this would probably be a lot more useful for the community than option 1, but would of course be more work).

Specific comments

In some cases, the manuscript methodology could benefit from further explanation. For example: • I was very confused about how PO4 was handled in the model. On P.4 l. 108 it is stated that, "The model is run in off–line mode like in the studies performed by Palmiéri et al. (2015), Guyennon et al. (2015), Ayache et al. (2015, 2016a, b) and Richon et al. (2017). PISCES passive biogeochemical tracers are transported using an advection–diffusion scheme..." What was meant by the model being run offline? Of the references above, only Guyennon et al. and Richon et al. looked at biogeochemical processes – the others looked at processes involving actual passive tracers that do not behave like nutrients in the real ocean. In Guyennon et

al., they said, "the coupling between the hydrodynamic and biogeochemical models is offline, i.e., biological retroaction on the physics is not taken into account" – but it appeared to me that biogeochemistry was prognostically calculated in that reference but not in this paper. Even if passive nutrient tracers follow deep-sea observations very well based on an offline model, how can one assess the biogeochemical changes caused by P deposition at the surface as the authors do here, if biogeochemistry is not calculated prognostically? Please clarify. In the Richon et al., 2017 text, this uncertainty was not discussed. Also, if P is a passive tracer, how can it affect Chl a as discussed in section 3.4? Please clarify this point in the text as well, and address any associated uncertainty and implications of the method in the text.

• On a related note, how exactly was surface PO4 related to Chl a in the model? I did not see this discussed, or any of the associated uncertainties.

• Section 3.3 and figure 5: Where does the referred-to surface PO4 data come from? From the model or from observations?

• Section 3.1: How was P deposition estimated from aerosol concentration observations? Was a deposition velocity assumed, and if so, what assumptions were used?

• The usage of the terms "total P" and "total phosphorus" in the manuscript are confusing. In most of the literature on atmospheric P deposition, the term total P indicates the sum of all phosphorus in any form (soluble or insoluble, organic or inorganic). On p. 6 l. 172, the authors state, "We investigate the impacts of each source of PO4 by performing two different simulations: "PDUST" and "PCOMB"; they include, respectively, natural dust only and combustion–generated aerosol only as atmospheric sources of PO4. We also performed a "Total P" simulation with the two sources included." Although it is not completely clear, here the authors seem to me to imply that total phosphorus is actually the sum of phosphate only from dust and combustion sources. On p4 l. 117, the term "total phosphorus" seems to imply the same thing. Then on page 6 line 187, the authors state, "We used the times series of total P measured at 9 different

stations over the Mediterranean from the ADIOS campaign (Guieu et al., 2010) and the soluble P measured at 2 stations in the South of France from the MOOSE campaign (de Fommervault et al., 2015)". Here the authors seem to distinguish between soluble and total P, as I would have otherwise expected. Elsewhere in the manuscript, the authors also use the term "atmospheric P" (which to me implies total phosphorus) to mean atmospheric soluble PO4. I suggest clarifying these different concepts, and using separate terms for each. Along those lines, I also suggest changing the title in Fig. 6 from "Total P" to something else.

• On a similar vein, P1 l.15: "We examine separately the different soluble phosphorus (PO4) sources..." Please keep in mind again that soluble phosphorus and PO4 are different things. Soluble P includes soluble organic P, which was not discussed much in this manuscript, except as a small note late in the paper in section 4. To avoid confusion, I recommend being clearer about this in the text.

• The authors talk about other sources of surface PO4 (e.g., riverine and oceanic via Gibraltar). Were these data obtained only from the model? Is there literature data with relevant information? If so, that information would be good to put in Table 2 for reference and discussion in section 3.2. If these data are not available, that would be worth mentioning and discussing.

My main concern, as mentioned, was that a few key uncertainties were either not made clear enough or fully addressed. These include:

1) Non-Redfieldian marine biogeochemical dynamics. The authors state on P. 4 l. 102 that: "PISCES is a Redfieldian model: the C/N/P ratio used for biology growth is fixed to 122/16/1." Many recent studies have discussed the shortcomings of this assumption in the real ocean, particularly in oligotrophic regions like the Mediterranean. A very large body of work shows that Redfield dynamics may be particularly erroneous with respect to P cycling (e.g., work by M. Lomas, R. Letscher, A. Landolfi, etc. (this is not a comprehensive list)). Given that Redfieldian assumptions are unlikely to represent

actual biogeochemical dynamics in this paper's study region, I feel that the authors must spend much more time discussing this uncertainty. It would be good if they could also more clearly state what meaningful information the results provide, given this large uncertainty. Ideally, they would also run additional model tests under non-Redfieldian assumptions.

2) The influence of soluble organic P in deposition was only touched upon in the manuscript. However, various studies suggest that it could be an important, or even dominant, source of soluble phosphorus to organisms in addition to the PO4 covered in this study (e.g., Chen et al., 1985; Kanakidou et al., 2012 and references therein). Particularly relevant for this paper is the fact that soluble organic P, in the few cases where it has been measured, appears to be much larger in combustion-sourced aerosols than in dust aerosols (e.g., Longo et al., 2014; Zamora et al. 2013). The authors should discuss the implications of/uncertainties related to not including organic P in their analysis. To make the paper more useful to the community, they may also consider running sensitivity tests estimating the potential impact on their results of including this additional P source.

3) Uncertainties with the model assumptions themselves require further discussion. For example: • The majority of the results focus and rely on modeled ocean surface PO4 concentrations. However, the majority of the model evaluation focuses on subsurface ocean PO4 trends, or surface Chl a trends. There was no in-depth discussion of how well the model compared to surface PO4 data, or what kind of data were available for this comparison. Moreover, the authors do not discuss how surface Chl a is related to surface PO4, either as parameterized in the model, or in actual observations. • Relatedly, on P11, l.342 the authors state: "Based on our large scale LMDz–INCA model, we estimate that combustion is responsible for 7 % on average of total PO4 supply. In comparison, the average contribution of Pdust to PO4 supply is 4 % (Table 2)." These are very precise numbers that imply high confidence. What is the certainty in the other P sources? Please rephrase, or discuss further.

4) Potential effects of nutrient co-limitation on the results. Most of the studies that I know of (although I am not an expert), indicate that phosphorus may be co-limiting along with other nutrient sources. This may also be worth discussing further.

I also had a variety of other, more minor suggestions/concerns:

P2l.27: "The most important aerosol deposition fluxes to the global ocean are induced by sea salt and natural desert dust (Goudie, 2006; Albani et al., 2015) respectively corresponding to material recycling and external inputs." Did the authors mean "most important" here (which is dependent on the process of interest) or something like, "largest by mass"? Please rephrase.

p.2 l. "It is especially important to constrain external sources of phosphorus because it limits productivity in many regions of the oceans." Reference?

p. 2 "The main sources of atmospheric phosphorus for the surface waters of the global ocean are desert dust, sea spray and combustion from anthropogenic activities (Graham and Duce, 1979; Mahowald et al., 2008). " I don't think sea spray should be considered a source, because as the authors stated, it is recycled material.

P2l54: "The Mediterranean Sea is also a hot–spot for climate change impacts (Lejeusne et al., 2010), in part because it is the recipient of aerosols from a variety of different geographical sources." I don't see how being the recipient of aerosols from a variety of geographical sources makes the Mediterranean Sea a hotspot for climate change impacts (was that referenced in the Lejeusne article somewhere)? Suggest rewording.

P.4 l. 95: "These evaluations showed satisfying results." Please be more specific?

p. 4, l. 111: "Biogeochemical characteristics of the latest version of the NEMOMED12/PISCES model are evaluated in Richon et al. (2017)." Am I correct in understanding that the Richon et al., 2017 model setup is very similar and relevant to this work? If so, I recommend that the authors just cite this paper and summarize

the relevant information on how well the model performs from Appendix A in the text, instead of including Appendix A which just repeats the information in Richon et al., 2017 as far as I can tell. Figures A1 and A2 are already in Richon et al., 2017 almost exactly, so those can also be removed.

Figure A2 (if you decide to keep it): Currently, it's hard to understand this figure since it is not clear what is east and west (although east and west are discussed in the referring text), and since the surface observations are unidentifiable in its present form. Please note the latitudes/longitudes of the points somehow (e.g., by having an insert with a cruise track). Please also label the x-axis (is this distance in m)? Please present the data in a different way so that the reader can see the nutricline information better (e.g., with an insert, following Richon et al., 2017, or by presenting the data on depth/nutrient plots). Again, I suggest just removing this figure entirely and referencing Richon et al., 2017.

P5, l.151: "Another important source of P aerosols in this region is sea spray" I recommend removing the word "source" and with something like "input" since recycled aerosols are not really a new source of P.

P7, l213: "The underestimation of total P deposition is also likely due in part to our omission of P from other potential sources such as PBAP and sea salt." Estimating deposition velocities from aerosols accurately is a major challenge (e.g., Jickells et al., 2017; Baker et al., 2017; Duce et al., 1991) and it is associated with high uncertainties in deposition fluxed to the ocean surface. I think this would be worth mentioning and keeping in mind as another major uncertainty for this comparison.

Figure 2 caption: please note somewhere that this is model output.

Table 2: Please mention in the Table or the caption that these estimates are model-derived. Also, as mentioned, the caption "Total P" is confusing –please clarify what you mean here – I think this value include riverine P? If so, please title this with something else distinguishable from total P in aerosols, and total sources of soluble PO4. Does

the Krom et al estimate include rivers? Please specify.

Fig. 4: Please define in the caption what the red and black bars indicate (which is where my eye goes first to find this information). Also, it would be useful to have the same numbers in the different regions that correspond to their label in Figure 2. Also, please clarify the units of the bar plots.

P.8, l. 244: "Our previous study showed that June is the period of most significant impacts from aerosol deposition in spite of the low fluxes, due to thermal stratification (Richon et al., 2017). " Please be more specific here - most significant impacts on what?

p.8, l. 248: "The North Adriatic is under strong influence of riverine inputs and atmospheric deposition of P from combustion (Figure 3)". Did you mean Fig.4?

Section 3.2: it might be useful (although not strictly necessary for me to recommend for publication) to know how your model dust observations compare with AOD trends in the region, which are available during your study period.

P9, l. 273: "Atmospheric phosphorus deposition has different impacts on PO4 concentration depending on the source, the location, and the period of the year." Suggest changing to, "Atmospheric phosphorus deposition has different impacts in the model on PO4 concentration depending on the source, the location, and the period of the year."

Section 3.3 and figure 5: Please define "maximal relative effects" and "relative impacts" and what a percent of average maximal relative effect means and how it is calculated. Where do you get the surface PO4 data? From the model or from observations? If in the model, how well does the model reproduce observations?

P9 l. 278: "Figure 5 shows the relative impacts of phosphorus deposition from the two sources (combustion and dust) on surface PO4 concentration for the month of June. The relative impacts of atmospheric deposition from different sources are varying over

time. . ." Please specify why you focus on June. You do not show or discuss how the relative impacts vary over time – please do so if you wish to keep this sentence.

Fig. 6: again, what does Total P represent in this instance? Pdust + Pcomb? Also, in the discussion of this figure, I think it is important to be much more focused on the uncertainties in your findings – e.g., regarding the relationship between modeled PO4 and Chl a, Redfieldian assumptions, etc.

P10 l.330: "We performed a Student's t–test on the grid matrix of relative impacts of Pdust and Pcomb over the three regions . . . and found that the mean values are statistically different (p–value < 0.01). This shows that even though the impacts of Pdust are close to the effects of Pcomb in the South Ionian, they are significantly dominant. " What do you mean by "dominant" specifically? Larger? Just because differences are significant, does not mean that the differences are meaningful. Please clarify (or remove the sentence, since it does not appear to be central to the paper).

P13, l.: "In the coastal Adriatic and Aegean Seas that are under strong influence of anthropogenic emissions, we showed that combustion-derived phosphorus deposition has effects on the biological productivity." Suggest rephrasing to: "In the coastal Adriatic and Aegean Seas that are under strong influence of anthropogenic emissions, we showed that combustion-derived phosphorus deposition may have effects on the biological productivity" or something similar. I also suggest emphasizing that your idealized experiment results indicate that these effects are likely to be fairly small, although other experiments with more realistic biogeochemistry are necessary to further constrain this problem.

Technical comments

P11, l. 372: "tshe" to "the"

  References

Baker, A. R., Kanakidou, M., Altieri, K. E., Daskalakis, N., Okin, G. S., Myriokefalitakis,

S., Dentener, F., Uematsu, M., Sarin, M. M., Duce, R. A., Galloway, J. N., Keene, W. C., Singh, A., Zamora, L., Lamarque, J.-F., Hsu, S.-C., Rohekar, S. S., and Prospero, J. M.: Observation- and model-based estimates of particulate dry nitrogen deposition to the oceans, Atmos. Chem. Phys., 17, 8189-8210, https://doi.org/10.5194/acp-17-8189-2017, 2017.

CHEN L.; ARIMOTO R.; DUCE R.A., 1985: The sources and forms of phosphorus in marine aerosol particles and rain from northern New Zealand. Atmospheric Environment 19(5): 779-788

Duce, R. A. et al. (1991), The atmospheric input of trace species to the world ocean, Global Biogeochem. Cycles, 5(3), 193–259, doi:10.1029/91GB01778.

Jickells, T. D., et al. (2017), A reevaluation of the magnitude and impacts of anthropogenic atmospheric nitrogen inputs on the ocean, Global Biogeochem. Cycles, 31, 289–305, doi:10.1002/ 2016GB005586.

Kanakidou, M., et al. (2012), Atmospheric fluxes of organic N and P to the global ocean, Global Biogeochem. Cycles, 26, GB3026, doi:10.1029/2011GB004277.

Longo, A. F., et al. (2014), P-NEXFS analysis of aerosol phosphorus delivered to the Mediterranean Sea, Geophys. Res. Lett., 41, 4043–4049, doi:10.1002/ 2014GL060555.

Zamora, L. M., J. M. Prospero, D. A. Hansell, and J. M. Trapp (2013), Atmospheric P deposition to the subtropical North Atlantic: sources, properties, and relationship to N deposition, J. Geophys. Res. Atmos., 118, 1546–1562, doi:10.1002/jgrd.50187.
* * *
**BGD**

Interactive
comment

---

## Author Comment (AC1) · 13 Oct 2017

We thank the editor and reviewers for their valuable review of our manuscript "Modeling the biogeochemical impact of atmospheric phosphate deposition from desert dust and combustion sources to the Mediterranean Sea". Below are our detailed responses to their questions and comments. The reviewer's comments are in *italic*, the author's answers in plain text, quotes from the manuscript are in quotation marks. We provide with this document a revised version of the manuscript and a track version allowing to visualizing changes from the submitted manuscript.

Jean-Claude Dutay published on July 25 on behalf of the authors a first reply to the reviewer 1: Referee1 is right to consider that having another set of forcings with higher resolution would represent an undeniable improvement for our modeling efforts. Unfortunately this simulation is not conceivable at short term, because such forcings are not available yet. For instance, the ALADIN-Climat model used in our previous study, presently does not simulate Phosphate from combustion (it has only Pdust at the moment), and this product will not be available soon since this development requires time. In order to make progress anyway on our scientific research, we were forced, for a preliminary study to use low resolution forcings, a classical strategy. We comment the limits of this approach in the manuscript and encourage for revisiting our conclusions with more refined forcings in the future. However we consider that our study has revealed some new interesting results, such as the spatial difference for the impact of Pcomb and Pdust and their relative quantification, that represent new information that deserve to be presented to our scientific community, as they have an importance for the modeling and the functioning of the biogeochemical cycle of the Mediterranean sea. We hope as well that this preliminary study will motivate atmospheric regional modeling group for producing more appropriate high resolution aerosols deposition field soon.

*The present manuscript "Modeling the biogeochemical impact of atmospheric phosphate deposition from desert dust and combustion sources to the Mediterranean Sea" proposes an analysis of the impact of phosphorus atmospheric deposition comparing different sources: namely desert dust (Pdust) and combustion sources (Pcomb). The idea is very interesting and useful because the two sources are, in principle, characterized by different spatial and temporal distributions. But, the main weakness of the manuscript is, in my opinion, the insufficient skill of the global atmospheric model LMDz-INCA in reproducing correctly the amount of dust deposition fluxes and its spatial and temporal variability for the Mediterranean area. Authors cite*

*another model, the higher resolution ALADIN-Climat (Nabat et al. 2012), used for a companion paper (Richon et al prog ocean. 2017), which gives higher deposition rate. I think that it is necessary to add also a test with the ALADIN-Climat model (equipped with the proper phosphorus deposition model), in order to have at least an ensemble composed by two members, this would make results more robust. Moreover, the choice of selecting only the year 2005 given the high variability of Pdust, is not clear to me. This high variability is important and its impact should be estimated. Therefore, I suggest that the present manuscript can be published only after major revisions of the simulation protocol.*

We thank the reviewer for these comments. As stated in the first reply by Jean-Claude Dutay (see also above), the use of the global model LMDz-INCA was driven by the availability of the model outputs. Indeed, we were not able to find any regional atmospheric model that treats the phosphorus cycle and distinguishes between phosphorus in dust and phosphorus from combustion. When we initiated this study, the only year for which P deposition from combustion had been computed was 2005, this guided our choice for simulating 2005 with NEMO/PISCES. The source of phosphorus from combustion has not yet been included in the ALADIN-Climat model and such development is not expected for a while. Considering these difficulties, we try to provide the best evaluation of deposition fluxes keeping in mind that both models and measurements have some uncertainties and that these induce uncertainties in our results.

We agree with the reviewer that it is difficult to conclude on the exact impact of atmospheric deposition on the biogeochemistry of the Mediterranean with such discrepancy between models and measurements. This is why we will in general try to emphasize more in the manuscript that the scope of this study is not to use models as predictors of the Mediterranean's functioning, but to test the sensitivity of an oligotrophic area such as the Mediterranean to contrasted atmospheric deposition patterns representative of some of the varied aerosol sources surrounding the Mediterranean. We hope that

these first results, along with their limitations, should encourage for more model and measurements development in the coming years.

In this paper, we point out the zonal distribution of atmospheric deposition from contrasted sources and the changes in the biogeochemistry of the different basins they impact. Therefore, even if the flux values are uncertain, there is a clear distinct distribution of phosphate deposition from the 2 considered sources. In this paper, we show that P deposition in the South of the basin (that is likely to come mainly from natural dust) has different impacts than P deposition in the North (that is likely to come from combustion sources).

We added to the discussion section some precisions about the scope and limitations of our study (lines 379-387) "The purpose of this study is to raise questions on the relative importance of the various aerosol sources that border the Mediterranean and their potential impacts on the nutrient supply and biological productivity of the basin. The literature on the Mediterranean aerosols is often centered on Saharan dust deposition which is believed to have the highest impact on the basin's biogeochemistry (e.g. Bergametti et al., 1992; Migon and Sandroni, 1999; Aghnatios et al., 2014). The study aims at shading new light on the other sources and their potential role, but, if Saharan dust does have an impact on the regional climate system and represents a source of particles (e.g. D'Almedia, 1986; Nabat et al., 2012), it may not be so dominant as previously believed as a source of bioavailable nutrients." And lines 402-406: "However, the underestimation of deposition fluxes shown by Figure 1 forces to consider that our results on the relative contributions of the different phosphate external sources are somewhat uncertain. More measurements and developments of the atmospheric modeling must be undertaken in order to make more precise assessments of the importance of atmospheric deposition as a source of nutrients to the oligotrophic Mediterranean."

*ABSTRACT lines 18-20: "The impact of the different sources of phosphate on the biogeochemical cycles is remarkably different and should be accounted for in*

*modeling studies." This sentence is, in my opinion, not clear, "remarkably different" with respect to what?* We agree with the reviewer and modified the sentence accordingly, to: "Differences in the geographical deposition patterns between phosphate from dust and the one from combustion will cause contrasted and significant changes in the biogeochemistry of the basin. These different sources should therefore be accounted for in modeling studies."

*Pg 4. line113:line 115 : "The model satisfyingly reproduces the vertical distribution of nutrients in the basin and the main productive zones that are the Alboran Sea, the Gulf of Lions and most coastal areas (see appendix)." The comparison/validation shown in the appendix appears quite subjective, no objective statistical indicators are provided.* It is complicated to produce reliable statistics over model simulations and difficult to find available data covering our entire simulation period/area. Therefore, with such sparse data, most of the validation in many modeling studies is only qualitative. We calculated the average and standard deviation of chlorophyll values measured and modeled at the DYFAMED station (Ligurian Sea) for the 1997-2005 period (see Richon et al 2017, Prog. Ocean.). We found that the average measured chlorophyll a in the top 200 meters is 0.290 $\pm$ 0.177 10-3 g m-3 and the average model value is 0.205 $\pm$ 0.111 10-3 g m-3. For PO4, the average measured value is 0.234 $\pm$ 0.085 mmol m-3 and the modeled average is 0.167 $\pm$ 0.179 mmol m-3. This has been added in the manuscript lines 129-133. It should be noted that as suggested by Reviewer 2, we withdrew the appendix because the model evaluation is already shown in Richon et al. (2017, Prog. Ocean.) and shows the same figures and results.

*Pg6, line 198: Pg 7, line 200: "We were able to compare the dust deposition flux modeled with LMDz–INCA used to derive Pdust deposition over the ADIOS sampling period with the measurements. The comparison is shown in Table 1. The dust fluxes produced by the model are realistic." I plotted results reported in Tab 1. See figure attached (x-axis stations, y-axis dust deposition, units are g m-2yr-1). In my*

*opinion dust fluxes produced by the model (brown line; MODEL "ADIOS period") com-
pared to data (blue line; DATA "ADIOS period") are very different. In particular there is
a strong underestimation of the model, about an order of magnitude, and the spatial
variability across stations is absent in the model. So the sentence "The dust fluxes
produced by the model are realistic", should be substituted by something like "model
presents a strong underestimation compared to data and it is not able to represent the
spatial variability of the data". Clearly, as stated by Authors, the dataset available is not
enough, and continuous time series at different stations should be used to corroborate
the model. But, given this situation, the usage of another atmospheric model, for
example the ALADIN-Climat is, in my opinion, mandatory. A higher resolution model
would allow for more robust results in terms of spatial gradients of dust deposition also.*
In Figure 1 of the article, we provide the comparison of total phosphorus deposition
(from dust and from combustion) from LMDz-INCA with total phosphorus deposition
measured at different stations. Although we stress that model and observation data
are not from the same year, this Figure helps evaluate the modeled fluxes against
the rare measurements we found in order to point out the uncertainties of the model.
In the Table 1, the dust deposition fluxes for the period 2001-2002 corresponding to
the ADIOS campaign are based on model outputs with a lower resolution ($1.27°$ in
latitude by $2.5°$ in longitude) than those for the year 2005 ($0.94°$ in latitude by $1.28°$ in
longitude). As stated by Bouet et al. (2012), dust emission (and hence its deposition)
is highly sensitive to model resolution. Therefore, the coarse resolution of the dust
model used in table 1 for 2001-2002 may explain the underestimation and the lack
of spatial variability from the model. We also noted that comparison appears better
(within a factor of 2) at the 4 stations of the eastern Mediterranean (Cyprus, Greek
Islands and Turkey). We added this precision in Section 3.1 (lines 234-243).

Comparison of LMDz-INCA phosphorus deposition fluxes to measurements on the
global scale are provided by Wang et al (2017) Global Change Biology (see Figure 2
below), it showed that the normalized bias observed at the global scale is coherent
with the underestimation we observe in the Mediterranean.

For comparison, we show in the following table taken from Richon et al. (2017, Prog. Oceanog.) the comparison of dust deposition from the ALADIN-Climat regional model with the measured values during the ADIOS campaign.

This Table illustrates that atmospheric deposition fluxes of mineral dust produced by ALADIN-Climat are higher than those from LMDz-INCA. However, the spatial variability of ALADIN is also low compared to point observations, and the fluxes are overestimated. These results seem to show that the 50 km resolution of the regional model ALADIN-Climat is still not enough to reproduce the spatial variability observed in the measurements. Moreover, the model ALADIN-Climat does not include sources of atmospheric P other than desert dust at the moment, which explains that it is not used in our present study.

*Pg 7, lines 221: lines 226: also in this case the estimates for the total deposition flux by the model seem low. In a recent paper [Powley et al. (2017), Global Biogeochem. Cycles, 31, 1010-1031] Authors report atmospheric deposition rates of 0.16 109 mol/yr WMS and of 0.38 109 mol/yr EMS (see their Tab. 3). In the present manuscript the estimates are much lower. Given the lack of data it is difficult to reach a conclusion, but anyway this discrepancy raises the question of how robust is the discussion on spatial gradients if the average values present such an uncertainty.* We thank the reviewer for this reference. Powley et al. use different estimations of total P deposition for their assessment, among which, the ADIOS campaign data included in our paper. The average values of Powley et al. over the basins are higher than our estimates. However, the measurements used include total P deposition (bulk deposition from all sources) whereas our model estimates do not include biogenic and volcanic sources. Moreover, the extrapolation of the deposition fluxes measured in a few localities to a basin scale average deposition flux as in Powley et al. (see their supplementary material) may lead to high uncertainties in the estimates. In particular, this method may not represent the important gradients in deposition from coastal to pelagic areas. We are conscious that neither the model, nor the measurements

can be representative of the full temporal and spatial variability of the fluxes, and that this variability is probably underestimated by both models and extrapolations of measurements. We agree, however, that is it important to consider all estimation methods in order to get a picture as precise as possible of atmospheric deposition over the Mediterranean.

*Pg 8, line 239 "However, riverine inputs are the dominant external source of phosphate for almost all Mediterranean regions" Given the uncertainty on phosphorus deposition, and apparently its underestimation, this sentence appears not demonstrated.* We modified this sentence to: "According to our model result, which remain highly uncertain, the riverine inputs computed from the PISCES model would constitute the main phosphate source to the Mediterranean Basin. They account for over 85

*Pg 11, line 363: line 365 : "The atmospheric model LMDz–INCA has a low resolution given the regional Mediterranean scale: Pdust deposition forcing has 280x193 grid points globally and 500 grid points covering the Mediterranean, and Pcomb forcing has 144x143 grid points in total and 200 grid points covering the Mediterranean. These forcings reproduce well the average deposition patterns at the basin scale but may not be reliable when analyzing small scale deposition patterns." The statement that the global forcing reproduces well the average deposition should be somehow proved.* In a recent article, Wang et al. (2017) compare the deposition fluxes of total phosphorus (from dust and combustion) from the LMDz-INCA model with measurements from all regions of the world. The following Figure (Figure 3c in their article) shows a good correlation between modeled and measured fluxes in spite of some underestimations in many regions, among which, the Mediterranean. We change the sentence to "These forcing reproduce realistic deposition patterns at the global scale, in spite of generally underestimating the measured fluxes..."

*Pg 12, line 381:line 384 : "Natural dust emissions, transport and deposition to the Mediterranean are shown to be highly variable from a year to the next (e.g. Moulin et al., 1997; Laurent et al., 2008; Vincent et al., 2016) so that the relative contributions of Pcomb and Pdust may also vary." Authors focused on an average (or median) year, namely the 2005. But given the high inter-annual variability, at least of Pdust, what is the meaning of such a choice? It would be better to consider many years and analyse the temporal variability to have a better quantification of the reality? What is the role of extremes?* As stated in the methods section, 2005 is the only available year for Pcomb deposition. We are conscious that reducing the simulation period to 1 year prevents us to conclude on inter annual variability effects. Naturally, conducting similar experiments with several deposition years would be necessary. Unfortunately, daily atmospheric deposition of combustion-derived phosphorus was only simulated for the year 2005. We consider that, given the high spatial and temporal variability of deposition fluxes, a monthly resolution of deposition, as available for other years (Wang et al. 2017, GCB) would be a too strong and unnecessary limitation in simulating the biological response. We added this sentence in the manuscript lines 151-154.

The monthly deposition of phosphorus from combustion has recently been simulated over the 1997-2012 period by Rong Wang. The following plot represents the yearly bioavailable phosphate deposition (in kg year-1) over the entire Mediterranean for the 1997-2012 period. In blue is the phosphate from combustion and in green the phosphate from dust. We can observe that 2005 is not an exceptional year in terms of deposition flux and that in spite of some inter-annual variability, mass deposition of phosphate from combustion seems to be, at the basin scale, always dominant over dust-derived phosphate deposition for the period at our disposal. We added a sentence in the discussion (line 390): "The relative dominance of combustion over dust as a source of phosphate for the Mediterranean seems to be confirmed by the analysis of yearly deposition fluxes of Pdust and Pcomb over the 1997-2012 period (not shown)."

Even though these deposition fluxes are only based on monthly values, we can

conclude that, on a yearly basis, combustion seems to be, according to this model, dominant over dust as a source of phosphate. However, the low temporal resolution of Pcomb deposition does not allow us to conclude on the importance of short term events such as Saharan dust outbreaks that may lead to local dominance of P deposition from dust.

Please also note the supplement to this comment:
https://www.biogeosciences-discuss.net/bg-2017-242/bg-2017-242-AC1-supplement.pdf

| Station | ADIOS | ALADIN (ADIOS period) | ALADIN (1982 - 2012) |
|---|---|---|---|
| Cap Spartel, Morocco | 6.8 (108) | 15 (135) | 19 (42) |
| Cap Béar, France | 11 (120) | 18 (113) | 15 (46) |
| Corsica, France | 28 (275) | 20 (116) | 19 (55) |
| Mahdia, Tunisia | 24 (127) | 62 (124) | 45 (52) |
| Lesbos, Greece | 6.0 (101) | 27 (115) | 42 (79) |
| Crete, Greece | 9.0 (199) | 24 (129) | 42 (78) |
| Akkuyu, Turkey | 10 (99) | 23 (119) | 26 (79) |
| Cavo Greco, Cyprus | 4.1 (63) | 27 (120) | 35 (80) |
| Alexandria, Egypt | 21 (74) | 30 (142) | 31 (77) |

Table 3: Dust deposition fluxes (g m$^{-2}$ yr$^{-1}$) measured during the ADIOS campaign (derived from Al measured deposition fluxes considering that dust contains 7 % of Al), simulated by the ALADIN-Climate model (June 2001 - May 2002) and values simulated by ALADIN-Climate model on the whole period available (1982–2012). Values in brackets indicate the relative standard deviations of monthly fluxes calculated as (standard deviation)*100/mean.

**Fig. 1.** Dust deposition fluxes from the ALADIN-Climat model compared to the ADIOS campaign measurments (Table from Richon et al. 2017, Prog. Ocean.)

(c)

All data

NMB = −.53

$R^2 = .25$ (116)

Modeled P dep. (mg m$^{-2}$ yr$^{-1}$)

Measured P dep. (mg m$^{-2}$ yr$^{-1}$)

**Fig. 2.** Comparison of modeled and measured P fluxes for the global LMDz-INCA model. Figure from Wang et al. (2017, Global Change Biology)

![Figure 3 plot showing total deposition of soluble phosphate]

**Fig. 3.** Total deposition of soluble phosphate from combustion (blue line) and natural dust (green line) over the entire Mediterranean basin for the 1997-2012 period.

**Supplement:**

[revised manuscript text omitted]
 nutrient fluxes from the Atlantic are computed as the product of nutrient concentrations in the buffer zone by the water fluxes through the Strait of Gibraltar computed by the model.

The model is run in off–line mode like in the studies performed by Palmiéri et al. (2015), Guyennon et al. (2015), Ayache et al. (2015, 2016a, b) and Richon et al. (2017). PISCES  biogeochemical tracers are transported using an advection–diffusion scheme driven by dynamical variables (velocities, pressure, mixing coefficients...) previously calculated by the oceanic model NEMO. Biogeochemical variables are prognostically calculated and not read from forcing files. Biogeochemical characteristics of the latest version of the NEMOMED12/PISCES model are evaluated in Richon et al. (2017). The model satisfyingly reproduces the vertical distribution of nutrients in the basin in spite of a too smooth nutricline, and the main productive zones that are the Alboran Sea, the Gulf of Lions and most coastal areas. Richon et al. (2017) calculated the average and

standard deviation of chlorophyll values measured and modeled at the DYFAMED station (Ligurian Sea) for the 1997–2005 period and found that the average measured chlorophyll a in the top 200 meters is $0.290 \pm 0.177 \ 10^{-3}$ g m$^{-3}$ and the average model value is $0.205 \pm 0.111 \ 10^{-3}$ g m$^{-3}$. For PO$_4$, the average measured value is $0.234 \pm 0.085 \ 10^{-3}$mol m$^{-3}$ and the modeled average is $0.167 \pm 0.179 \ 10^{-3}$mol m$^{-3}$. We report in appendix additional figures on the comparison of modeled and measured surface PO$_4$ concentrations.

The model NEMOMED12/PISCES is run in the same configuration than in Richon et al. (2017) who provide an evaluation of the model. In particular, the authors show that NEMOMED12/PISCES reproduces a correct west–to–east gradient of productivity when compared to satellite chlorophyll estimates in spite of some underestimation in the areas of high productivity such as the Gulf of Lions that they trace back to the circulation anomalies in the western basin. The vertical distribution of nutrients is satisfyingly reproduced by the model, in spite of underestimations in the Levantine Intermediate Waters (LIW) because of the too smooth nutricline.

[revised manuscript text omitted]

---

## Author Comment (AC2) · 13 Oct 2017

*General comments:*
*In this work, the authors assess how modeled phosphate deposition output from dust and combustion aerosols can affect the phosphate fluxes into the surface waters of the Mediterranean Sea. The oligotrophic Mediterranean is phosphorus stressed, limited, or co-limited in certain regions/species, and atmospheric deposition may be an important source of this nutrient. Given high anthropogenic impact on aerosols in this region, and potential future enhancements in surface water stratification, this is*

[Figure]

*a topic worthy of study. The methodology in this paper was good in most cases, and some of the important uncertainties were discussed very thoroughly. I have pointed out in the specific comments several places where the manuscript requires further explanation of the methodology. My main issue is that, in my opinion, the importance of this study was overstated, and that a few key uncertainties in the findings were downplayed too much (e.g., nutrient co-limitation, the influence of soluble organic P in deposition, non-Redfieldian marine biogeochemical dynamics, and some important model uncertainties). Because of this latter concern, I suggest the authors proceed in one of two ways: 1) Scale back the conclusions substantially, to focus on the differences between model estimated Pcomb and Pdust deposition and their potential implications in a (more clearly-emphasized) highly-simplified Redfieldian ocean, or 2) Maintain the scope that the authors do now, but also present results from non-Redfield experiments with prognostic biogeochemistry (this would probably be a lot more useful for the community than option 1, but would of course be more work). We thank the reviewer for these comments. It is true that the Mediterranean is likely to be a non-Redfieldian Sea and modeling this behavior would give interesting insights. However, the present version of the biogeochemical model PISCES we use is in a redfieldian configuration. A new version of the PISCES model is being developed with non-redfieldian ratios (Aumont, in prep). This non-redfieldian version of PISCES has been developed to treat the global ocean, and qualifying it for the Mediterranean Sea will likely take a few years more to come up with satisfactory results for the region. Specific comments In some cases, the manuscript methodology could benefit from further explanation. For example: I was very confused about how PO4 was handled in the model. On P.4 l. 108 it is stated that, "The model is run in off–line mode like in the studies performed by Palmiéri et al. (2015), Guyennon et al. (2015), Ayache et al. (2015, 2016a, b) and Richon et al. (2017). PISCES passive biogeochemical tracers are transported using an advection–diffusion scheme. . ." What was meant by the model being run offline? Of the references above, only Guyennon et al. and Richon et al. looked at biogeochemical processes – the others looked at processes involving*

*actual passive tracers that do not behave like nutrients in the real ocean. In Guyennon et al., they said, "the coupling between the hydrodynamic and biogeochemical models is offline, i.e., biological retroaction on the physics is not taken into account" – but it appeared to me that biogeochemistry was prognostically calculated in that reference but not in this paper. Even if passive nutrient tracers follow deep-sea observations very well based on an offline model, how can one assess the biogeochemical changes caused by P deposition at the surface as the authors do here, if biogeochemistry is not calculated prognostically? Please clarify. In the Richon et al., 2017 text, this uncertainty was not discussed. Also, if P is a passive tracer, how can it affect Chl a as discussed in section 3.4? Please clarify this point in the text as well, and address any associated uncertainty and implications of the method in the text.*

We changed the following sentence that brought confusion about the way PISCES treats biogeochemical tracers: "PISCES passive biogeochemical tracers are transported using an advection–diffusion scheme." Into "PISCES biogeochemical tracers are transported using an advection–diffusion scheme "... In PISCES, PO4 is one of the nutrients necessary for plankton growth; it is not a passive tracer. Phosphate concentration, as well as the 4 other nutrients represented in PISCES (NO3, NH4, Si and Fe), are used to calculate the nutrient limitation terms (that have a Michaelis-Menten formulation). These limitation terms allow calculating the productivity terms based on the use of each nutrient. The phytoplankton biomass, which is linked to chlorophyll production, is then derived from the productivity terms. All equations are in Aumont et al. (2015).

Offline models, in contrast to online or coupled models are run thanks to the use of climatological values of physical and biogeochemical boundary fluxes. In our case, the physics of the ocean is described by the model NEMO, and the biogeochemical cycles are represented by the PISCES model. NEMO allows calculating the movements of water masses using climatological values (forcings) of atmospheric and physical conditions such as winds, runoff or precipitations. The biogeochemical model PISCES calculates the biogeochemical state of the Mediterranean (nutrient and tracers concentrations) using the physical state from NEMO and biogeochemical conditions from climatologies such as nutrient inputs from rivers. The biogeochemistry in PISCES is calculated in the same way as in Guyennon et al. To clarify, we add the sentence "Biogeochemical variables are prognostically calculated and not read from forcing files."

*On a related note, how exactly was surface PO4 related to Chl a in the model? I did not see this discussed, or any of the associated uncertainties.*
To clarify, we added in section 2.1 "The concentration of nutrients is linked with phytoplankton productivity and chlorophyll-a production according to the equations described in Aumont et al. (2015). Phytoplankton growth rate is dependent on nutrient concentrations via the growth limiting factors"

*Section 3.3 and figure 5: Where does the referred-to surface PO4 data come from? From the model or from observations?*
We changed the Figure 5 legend "Map of daily maximal relative effects of total (Pdust + Pcomb) deposition in June 2005 on the surface phosphate concentration (0—10 m) compared to the reference simulation without atmospheric P deposition."

*Section 3.1: How was P deposition estimated from aerosol concentration observations? Was a deposition velocity assumed, and if so, what assumptions were used?*
In the LMDz-INCA model, an explicit deposition scheme is implemented. It represents 3 physical processes of deposition including sedimentation, turbulent dry deposition and wet deposition (in-cloud and below-cloud scavenging). These schemes allow accounting for more complex physical processes than the simple hypothesis of a constant deposition speed.
The observations we use for model evaluation are direct measurements of phosphorus bulk deposition (see Guieu et al. 2010 mar. chem. for protocol details). No estimations from atmospheric concentrations or optical properties such as AOD are used.

*The usage of the terms "total P" and "total phosphorus" in the manuscript are confusing. In most of the literature on atmospheric P deposition, the term total P indicates the sum of all phosphorus in any form (soluble or insoluble, organic or inorganic). On p. 6 l. 172, the authors state, "We investigate the impacts of each source of PO4 by performing two different simulations: "PDUST" and "PCOMB"; they include, respectively, natural dust only and combustion–generated aerosol only as atmospheric sources of PO4. We also performed a "Total P" simulation with the two sources included." Although it is not completely clear, here the authors seem to me to imply that total phosphorus is actually the sum of phosphate only from dust and combustion sources. On p4 l. 117, the term "total phosphorus" seems to imply the same thing. Then on page 6 line 187, the authors state, "We used the times series of total P measured at 9 different stations over the Mediterranean from the ADIOS campaign (Guieu et al., 2010) and the soluble P measured at 2 stations in the South of France from the MOOSE campaign (de Fommervault et al., 2015)". Here the authors seem to distinguish between soluble and total P, as I would have otherwise expected. Elsewhere in the manuscript, the authors also use the term "atmospheric P" (which to me implies total phosphorus) to mean atmospheric soluble PO4. I suggest clarifying these different concepts, and using separate terms for each. Along those lines, I also suggest changing the title in Fig. 6 from "Total P" to something else.*

We agree with the reviewer that the use of different terms is quite confusing. We modified some sentences in the text to clarify this point:

Section 2.3: "From now on, we name "total P" the sum of bioavailable phosphate from dust and combustion (Pdust + Pcomb)."

Section 3.1 "We used the times series of total phosphorus measured at 9 different stations over the Mediterranean from the ADIOS campaign (Guieu et al 2010) and the soluble phosphate measured in the deposition at 2 stations in the South of France from the MOOSE campaign"

Section 3.3 We renamed the section "Impacts of atmospheric deposition on marine surface phosphate budgets ", and line 298: "Atmospheric deposition of phosphate

aerosols has different impacts"

"Figure 5 shows the relative impacts of phosphate deposition from the two sources (combustion and dust) on surface PO4 concentration for the month of June 2005. The relative impacts of atmospheric deposition from different sources are dependent on both the underlying phosphate concentration and the bioavailable phosphate deposition flux."

We use the term "phosphate deposition" in the text because our focus is on the deposition of this bioavailable nutrient.

*On a similar vein, P1 l.15: "We examine separately the different soluble phosphorus (PO4) sources. . ." Please keep in mind again that soluble phosphorus and PO4 are different things. Soluble P includes soluble organic P, which was not discussed much in this manuscript, except as a small note late in the paper in section 4. To avoid confusion, I recommend being clearer about this in the text.*

We thank the reviewer for this remark. We are conscious that soluble P can also describe organic P. In this manuscript, we refer to the only soluble phosphorus form in PISCES which is PO4. We replace soluble phosphorus in this sentence by "phosphate".

*The authors talk about other sources of surface PO4 (e.g., riverine and oceanic via Gibraltar). Were these data obtained only from the model? Is there literature data with relevant information? If so, that information would be good to put in Table 2 for reference and discussion in section 3.2. If these data are not available, that would be worth mentioning and discussing.*

As described in section 2.1, riverine fluxes of nutrients are prescribed from Ludwig et al. 2009. This study groups the nutrient fluxes from 239 rivers around the Mediterranean and Black Sea obtained from measurements and model data. Unfortunately, the estimations of riverine fluxes are not available after 2000. This is why we used the riverine fluxes from the year 2000 in our study. The nutrient fluxes from the Atlantic are

computed as the product of the buffer zone concentrations constructed from the World Ocean Atlas (2005) and the water fluxes through the Strait of Gibraltar computed by the model. We added these precisions to the section 2.1.
*My main concern, as mentioned, was that a few key uncertainties were either not made clear enough or fully addressed. These include: 1) Non-Redfieldian marine biogeochemical dynamics. The authors state on P. 4 l. 102 that: "PISCES is a Redfieldian model: the C/N/P ratio used for biology growth is fixed to 122/16/1." Many recent studies have discussed the shortcomings of this assumption in the real ocean, particularly in oligotrophic regions like the Mediterranean. A very large body of work shows that Redfield dynamics may be particularly erroneous with respect to P cycling (e.g., work by M. Lomas, R. Letscher, A. Landolfi, etc. (this is not a comprehensive list)). Given that Redfieldian assumptions are unlikely to represent actual biogeochemical dynamics in this paper's study region, I feel that the authors must spend much more time discussing this uncertainty. It would be good if they could also more clearly state what meaningful information the results provide, given this large uncertainty. Ideally, they would also run additional model tests under non-Redfieldian assumptions.*

We thank the reviewer for this comment and refer to the general comments at the beginning of this section.

We added a paragraph on the implications of the use of a redfieldian model in the discussion section. "The PISCES version used in this study is based on the Redfield hypothesis that C/N/P ratios in organic cells are fixed. This fixed value determines the nutrient ratio for uptake and has the advantage of simplifying calculations in the 3-D high resolution coupled model. However, because the Mediterranean is highly oligotrophic, the biogeochemical cycles may be determined by non-Redfieldian dynamics (see Ribera d'Alcala et al. 2003, JGR). This non-Redfieldian behavior may imply complex nutrient limitations and co-limitations processes that can not be studied with the present PISCES version. To this day, there is no version of PISCES available that

includes the non-Redfieldian biogeochemistry in the Mediterranean. The development and use of such a version of PISCES is a perspective of this work that needs to be undertaken in order to fully understand nutrient dynamics and growth limitation process in the Mediterranean. This study provides first results on the potential impacts of phosphate atmospheric deposition on the Mediterranean nutrient pool and potential implications on biological productivity assuming the Redfield hypothesis."

*2) The influence of soluble organic P in deposition was only touched upon in the manuscript. However, various studies suggest that it could be an important, or even dominant, source of soluble phosphorus to organisms in addition to the PO4 covered in this study (e.g., Chen et al., 1985; Kanakidou et al., 2012 and references therein). Particularly relevant for this paper is the fact that soluble organic P, in the few cases where it has been measured, appears to be much larger in combustion-sourced aerosols than in dust aerosols (e.g., Longo et al., 2014; Zamora et al. 2013). The authors should discuss the implications of/uncertainties related to not including organic P in their analysis. To make the paper more useful to the community, they may also consider running sensitivity tests estimating the potential impact on their results of including this additional P source.*

We agree with the reviewer that including organic phosphorus is an important step for the community. Our hypotheses concerning phosphorus combustion in this study are only based on Mahowald et al (2008). However, Myriokefalitakis et al (2016) consider that organic phosphorus (DOP) can be deposited in the Mediterranean with combustion and biogenic aerosol. DOP is not included in the version of PISCES used in this study. However, if we consider the hypothesis of Kanakidou et al and Myriokefalitakis et al, and given that dissolved organic matter is recycled into inorganic nutrients in the sea, we may be able to consider the inclusion organic phopshorus as a source of atmospheric phosphate. We add some elements in the discussion section: "In the Mediterranean region that is surrounded by many forested areas, biogenic emissions may be an important source of atmospheric phosphorus in the form of

organic matter. Moreover, Kanakidou et al. (2012) show that an important fraction of organic phosphorus can be emitted from combustion. In particular, the numerous forest fires occurring every summer in the Mediterranean region may constitute an important source of organic phosphorus. However, the PISCES version used in this study does not include organic phosphorus. In the ocean, organic phosphorus can be recycled by bacterial activity into inorganic phosphate that is bioavailable for plankton growth. Therefore, the inclusion of organic phosphorus in PISCES along with an estimation of organic phosphorus from atmospheric fluxes is a perspective to consider."

*3) Uncertainties with the model assumptions themselves require further discussion. For example: The majority of the results focus and rely on modeled ocean surface PO4 concentrations. However, the majority of the model evaluation focuses on subsurface ocean PO4 trends, or surface Chl a trends. There was no in-depth discussion of how well the model compared to surface PO4 data, or what kind of data were available for this comparison. Moreover, the authors do not discuss how surface Chl a is related to surface PO4, either as parameterized in the model, or in actual observations.*

Figure 1 bellow displays the PO4 concentration on the BOUM section in the top 200 m (zoom from the A1 Figure from the manuscript). We have included in appendix a couple of figures to evaluate surface PO4.

BOUM is the most complete dataset available for the Mediterranean because it covers a full, recent west-to-east transect. There are very few estimations of nutrient concentration (and especially phosphate) in the surface layer of the Mediterranean (first 5-100 m) because the concentrations are so low that measures are often below the detection limit of sensors. In this figure, we can see that the model reproduces the increase in concentration below 50 m observed in the western basin but that the increase in concentration modeled in the eastern basin (below 100-150 m) is not observed.

For further comparison, we show in the figures below the average phosphate profiles

in different regions of the Mediterranean compared with data from Manca et al. (2004). These figures show that the model can reproduce the phosphate vertical distribution and that the model values are generally in the range of data standard deviations in surface waters.

*Relatedly, on P11, l.342 the authors state: "Based on our large scale LMDz–INCA model, we estimate that combustion is responsible for 7 % on average of total PO4 supply. In comparison, the average contribution of Pdust to PO4 supply is 4 % (Table 2)." These are very precise numbers that imply high confidence. What is the certainty in the other P sources? Please rephrase, or discuss further.*

The contribution values given in p.11, l342 are the values from Table 2. They are based on our modeling values and take into account only the sources of phosphate that are included in the simulations (namely rivers, Atlantic inputs, desert dust and combustion derived atmospheric phosphate). These are estimates for our present simulation that do not represent the absolute truth on the contribution of atmospheric phosphate deposition, but give light on the relative importance of the 2 atmospheric sources under the specific conditions of the year 2005, according to the LMDz-INCA model outputs. The purpose of this Table (and of this study in general) is to raise questions on the relative importance of the various aerosol sources that border the Mediterranean and their potential impacts on the nutrient supply and biological productivity of the basin. The literature on the Mediterranean aerosols is often centered on Saharan dust deposition which is believed to have the highest impact on the basin's biogeochemistry. The Table aims at shading new light on the other sources and their potential role. Acknowledging that model limitations makes those number highly uncertain, they suggest that Saharan dust might not be as dominant as it was previously believed as a source of bioavailable nutrients. We added these precisions in the discussion section. (lines 371-390).

*4) Potential effects of nutrient co-limitation on the results. Most of the studies*

*that I know of (although I am not an expert), indicate that phosphorus may be co-limiting along with other nutrient sources. This may also be worth discussing further.*
Reviewer is right to point out that nutrient co-limitation is a key question in the study of marine biogeochemical cycles and in particular in oligotrophic areas such as the Mediterranean. PISCES is a Monod type model in which nutrient limitations are calculated in a Michaëlis-Menten formulation. This means that growth rates of phyto-plankton increase linearly with nutrient concentrations when these concentrations are below a threshold. In an oligotrophic region such as the Mediterranean the concentra-tions are low enough for the growth rate to increase linearly with concentration. As a consequence, having no increase in productivity (that is linked to nutrient limitations) after phosphate deposition is a sign that growth rates are limited by at least one other nutrient (most probably N). We added a paragraph on section 3.4. "In general, we can identify 3 different biogeochemical responses in the 3 framed areas of Figure 7. Our hypothesis is that the different responses are linked to nutrient limitations. In the North Adriatic, the influence of coastal nutrient inputs leads to low nutrient limitation and high productivity. In the South Adriatic, the high impact of atmospheric phosphate deposition may be the sign of important phosphate limitation. Finally, the lack of response in South Ionian in spite of the relatively high atmospheric phosphate deposition probably indicates that the region is co-limited in P and N."

*I also had a variety of other, more minor suggestions/concerns: P2l.27: "The most important aerosol deposition fluxes to the global ocean are induced by sea salt and natural desert dust (Goudie, 2006; Albani et al., 2015) respectively corresponding to material recycling and external inputs." Did the authors mean "most important" here (which is dependent on the process of interest) or something like, "largest by mass"? Please rephrase.* Largest by mass.

*p.2 l. "It is especially important to constrain external sources of phosphorus be-cause it limits productivity in many regions of the oceans." Reference?*

We added a reference to Moore et al (2013).

*p. 2 "The main sources of atmospheric phosphorus for the surface waters of the global ocean are desert dust, sea spray and combustion from anthropogenic activities (Graham and Duce, 1979; Mahowald et al., 2008). I don't think sea spray should be considered a source, because as the authors stated, it is recycled material.*
We agree with the reviewer and removed sea spray.

*P2l54: "The Mediterranean Sea is also a hot–spot for climate change impacts (Lejeusne et al., 2010), in part because it is the recipient of aerosols from a variety of different geographical sources." I don't see how being the recipient of aerosols from a variety of geographical sources makes the Mediterranean Sea a hotspot for climate change impacts (was that referenced in the Lejeusne article somewhere)? Suggest rewording.*
We rephrased this part : "The Mediterranean Sea is also a hot-spot for climate change impacts. Moreover, it is the recipient of aerosols from a variety of different geographical sources. The impacts of aerosol deposition on the Mediterranean region are not fully understood and they may change in the future as a result of climate change impacts on land and sea."

*P.4 l. 95: "These evaluations showed satisfying results." Please be more specific?*
Changed to "These evaluations showed that the NEMO model is able to produce satisfying results when studying characteristics such as age-tracer of water masses of passive tracer transport."

*p. 4, l. 111: "Biogeochemical characteristics of the latest version of the NEMOMED12/PISCES model are evaluated in Richon et al. (2017)." Am I correct in understanding that the Richon et al., 2017 model setup is very similar and*

*relevant to this work? If so, I recommend that the authors just cite this paper and summarize the relevant information on how well the model performs from Appendix A in the text, instead of including Appendix A which just repeats the information in Richon et al., 2017 as far as I can tell. Figures A1 and A2 are already in Richon et al., 2017 almost exactly, so those can also be removed.*

The model setup in this paper is the same as the one in Richon et al (2017). In the present study, we compare the model outputs with data for the year 2005. We decided to follow the advice of the reviewer and removed the appendix. We added in section 2.1 "The model NEMOMED12/PISCES is run in the same configuration than in Richon et al. (2017) who provide an evaluation of the model. In particular, the authors show that NEMOMED12/PISCES reproduces a correct west-to-east gradient of productivity when compared to satellite chlorophyll estimates in spite of some underestimation in the areas of high productivity such as the Gulf of Lions that they trace back to the circulation anomalies of the western basin. The vertical distribution of nutrients is satisfyingly reproduced by the model in spite of underestimations in the Levantine Intermediate Waters (LIW) because of the too smooth nutricline."

*P5, l.151: "Another important source of P aerosols in this region is sea spray" I recommend removing the word "source" and with something like "input" since recycled aerosols are not really a new source of P.* Changed.

*P7, l213: "The underestimation of total P deposition is also likely due in part to our omission of P from other potential sources such as PBAP and sea salt." Estimating deposition velocities from aerosols accurately is a major challenge (e.g., Jickells et al., 2017; Baker et al., 2017; Duce et al., 1991) and it is associated with high uncertainties in deposition fluxed to the ocean surface. I think this would be worth mentioning and keeping in mind as another major uncertainty for this comparison.*

We agree with the reviewer that calculating deposition fluxes from aerosol concentration can lead to high uncertainties, in particular when extrapolating the fluxes to an

entire region based on average concentrations. This is the reason we tried to evaluate directly the modeled deposition fluxes, taking into account dry and wet deposition processes and daily variability.

Deposition time series are only available at a few stations, and observations are not available in 2005 (our model year). This leads to a high uncertainty in our comparison data. But we believe that these measurements are more reliable to compare deposition fluxes than basin scale estimations that do not account for large deposition gradients. Our approach is more process-based and should lead to less uncertainty than basin scale extrapolation from velocity fluxes. However, the exclusion of soluble organic phosphorus, PBAB and sea salt inevitably leads to some additional uncertainties.

*Figure 2 caption: please note somewhere that this is model output.*
Done

*Table 2: Please mention in the Table or the caption that these estimates are modelderived. Also, as mentioned, the caption "Total P" is confusing –please clarify what you mean here – I think this value include riverine P? If so, please title this with something else distinguishable from total P in aerosols, and total sources of soluble PO4. Does the Krom et al estimate include rivers? Please specify*
Precisions added in the figure caption

*Fig. 4: Please define in the caption what the red and black bars indicate (which is where my eye goes first to find this information). Also, it would be useful to have the same numbers in the different regions that correspond to their label in Figure 2. Also, please clarify the units of the bar plots.* Caption changed.

*P.8, l. 244: "Our previous study showed that June is the period of most significant impacts from aerosol deposition in spite of the low fluxes, due to thermal stratification (Richon et al., 2017). " Please be more specific here - most significant*

*impacts on what?*
Changed to "more important impacts on surface marine productivity"

*p.8, l. 248: "The North Adriatic is under strong influence of riverine inputs and atmospheric deposition of P from combustion (Figure 3)". Did you mean Fig.4?* We changed the reference

*Section 3.2: it might be useful (although not strictly necessary for me to recommend for publication) to know how your model dust observations compare with AOD trends in the region, which are available during your study period.*
We did not follow this reviewer suggestion. However, as previously stated, we do not believe that AOD is a good proxy for deposition. High deposition is generally related to rain, which means very cloudy conditions unfavorable to AOD measurements.

*P9, l. 273: "Atmospheric phosphorus deposition has different impacts on PO4 concentration depending on the source, the location, and the period of the year." Suggest changing to, "Atmospheric phosphorus deposition has different impacts in the model on PO4 concentration depending on the source, the location, and the period of the year."* Done

*Section 3.3 and figure 5: Please define "maximal relative effects" and "relative impacts" and what a percent of average maximal relative effect means and how it is calculated. Where do you get the surface PO4 data? From the model or from observations? If in the model, how well does the model reproduce observations?*
We rephrased the caption.

*P9 l. 278: "Figure 5 shows the relative impacts of phosphorus deposition from the two sources (combustion and dust) on surface PO4 concentration for the month of June. The relative impacts of atmospheric deposition from different sources are*

*varying over time. . ." Please specify why you focus on June. You do not show or discuss how the relative impacts vary over time – please do so if you wish to keep this sentence.*

As stated in section 3.2, we focus on June because it is the month of the year when maximal effects of deposition on surface productivity are observed. In Richon et al. 2017, we link this result to the vertical stratification and high surface nutrient limitations associated with sufficient deposition fluxes. We removed the term "varying over time" from the sentence because it was confusing.

*Fig. 6: again, what does Total P represent in this instance? Pdust + Pcomb? Also, in the discussion of this figure, I think it is important to be much more focused on the uncertainties in your findings – e.g., regarding the relationship between modeled PO4 and Chl a, Redfieldian assumptions, etc.*

We added the following information: "In this Redfieldian version of PISCES, chlorophyll production is linked with nutrient uptake that is constrained by the Redfield ratio. Therefore, the addition of excess nutrient will enhance chlorophyll production as long as other nutrients are bioavailable in the Redfield proportions. These results may change in a non Redfieldian model."

*P10 l.330: "We performed a Student's t–test on the grid matrix of relative impacts of Pdust and Pcomb over the three regions . . . and found that the mean values are statistically different (p–value < 0.01). This shows that even though the impacts of Pdust are close to the effects of Pcomb in the South Ionian, they are significantly dominant. " What do you mean by "dominant" specifically? Larger? Just because differences are significant, does not mean that the differences are meaningful. Please clarify (or remove the sentence, since it does not appear to be central to the paper).*

Reviewer is right. Also, given the high uncertainty on deposition fluxes, we chose to remove this sentence.

*P13, l.: "In the coastal Adriatic and Aegean Seas that are under strong influence of anthropogenic emissions, we showed that combustion-derived phosphorus deposition has effects on the biological productivity." Suggest rephrasing to: "In the coastal Adriatic and Aegean Seas that are under strong influence of anthropogenic emissions, we showed that combustion-derived phosphorus deposition may have effects on the biological productivity" or something similar. I also suggest emphasizing that your idealized experiment results indicate that these effects are likely to be fairly small, although other experiments with more realistic biogeochemistry are necessary to further constrain this problem.*

We thank the reviewer for this suggestion. We modified the sentence and added to this paragraph "In general, results from this idealized study suggest that the impacts of atmospheric deposition of phosphate are likely to be fairly small, even though atmospheric sources of phosphate seem to be important contributors to the total nutrient pool in some regions of the basin."
* * *
**PO4 BOUM Section**

PO4 mmol/m3

**Fig. 1.** PO4 concentration along the BOUM section (Moutin et al 2012). Zoom from the top 200m.

**vertical profiles PO4 2005 ave Adriatic South**

Depth vs PO4 mmol/m3 plot.

**Fig. 2.** Average concentration in 2005 in the South Adriaticregion (see map). Measurements and standard deviations are in pink, modeled values are represented by the green line.

[Figure]

Vertical profiles PO4 2005 and Alboran Sea

**Fig. 3.** Average concentration in 2005 in the Alboran Sea region (see map). Measurements and standard deviations are in pink, modeled values are represented by the green line.

[Figure]

**Fig. 4.** Average concentration in 2005 in the Algerian current region (see map). Measurements and standard deviations are in pink, modeled values are represented by the green line.

---

## Author Response (AR2)

**Responses to reviewer 1.**

*Review of the manuscript "Modeling the biogeochemical impact of atmospheric phosphate deposition from desert dust and combustion sources to the Mediterranean Sea" by Richon et al.*
*General comments*
*The present manuscript "Modeling the biogeochemical impact of atmospheric phosphate deposition from desert dust and combustion sources to the Mediterranean Sea" proposes an analysis of the impact of phosphorus atmospheric deposition comparing different sources: namely desert dust (Pdust) and combustion sources (Pcomb).*
*As I stated previously in my opinion the idea is very interesting and useful because the two sources are, in principle, characterized by different spatial and temporal distributions.*
*In my opinion still remains the problem of the uncertainty related to the coarse resolution of the LMDz-INCA global model therefore an effort to give some sort of quantification of such uncertainty in the results would be useful.*
*Nonetheless given the fact that Authors declare that it is not possible to integrate the simulations with a proper high resolution model (e.g. ALADIN-Climat) in my opinion the present work represents what is currently possible to perform. A possible way to give some estimates of the inter-annual variability would be to consider the monthly mean forcings available.*

**Specific comments**

*Pg 4 line 120: "The estimations of riverine fluxes are not available after 2000. Therefore, we use the riverine fluxes from the year 2000 in our study."*

*Why not using at least the 1995-2000 average value ? Is year 2000 representative?*
We chose to repeat the 2000 river forcing because it is the method used in other studies performed with the same model (see Richon *et al.* 2017, prog. Ocean.).
The 1995-2000 average river discharge of PO4 and NO3 are respectively 796 ± 279 ktPO4/year and 11500 ± 2100 ktNO3/year. The 2000 values we use are respectively 456 ktPO4/year and 9820 ktNO3/year. The NO3 value is in the variability range of the 1995-2000 period. The PO4 value is slightly below this range. However, data from Ludwig *et al.* (2009) indicate a decreasing trend of PO4 river discharge over the Mediterranean that this value is representative of.

*Pg 5 line 161-164: "We considered that given the high spatial and temporal variability of atmospheric deposition fluxes, a monthly resolution of deposition, as available for other years (Wang et al., 2017), would be a too strong and unnecessary limitation in simulating the biogeochemical response."*

*In my opinion given the fact that not all the results are discussed at high (daily) frequency in the present manuscript, also monthly forcings (that, if I understand correctly, are available for other years than 2005) could be acceptable and therefore used to carry on a multi-year simulation and obtain more solid results. I would suggest to keep the 2005 year simulation, with high frequency forcings but also to add, if technically possible, a simulation with monthly fields in a multi year simulation framework. For example looking at Table 2 the dust deposition of the ADIOS period simulated by the LMDz-INCA appears quite different from the LMDz-INCA (2005) estimates, so it appears worthy to consider also the inter-annual variability, in order to evaluate the variance.*
We provide in Figure 2 in the article the annual variation of Pcomb deposition over the Mediterranean for the 1997-2012 period derived from monthly deposition fields. This Figure shows that the inter-annual variability of Pcomb deposition is low.

[Figure]

*Figure 1: Inter-annual average Pcomb deposition over the Mediterranean*

Also, Rong Wang (pers. Comm.) computed the average deposition, monthly standard deviation and standard deviation to average ratio of Pcomb deposition based on monthly forcings for the 1997-2012 period (Figure 3 of the article and hereafter). These figures show that, over the Mediterranean, deposition variability is very low. Therefore, these results seem to confirm that 2005 is not an exceptional year for Pcomb deposition.

We added these figures in the article with the following lines (197-202) : « In order to assess inter-annual variability of the *Pcomb* deposition, Figure 2 shows the annual average *Pcomb* deposition over the Mediterranean for the 1997-2012 period. Figure 3 shows the standard deviation to average ratio of *Pcomb* deposition over the Mediterranean computed from LMDz-INCA for the 1997-2012 period. These figures show that *Pcomb* deposition has low inter-annual variability. Therefore, 2005 can be considered as a study year because both *Pcomb* and *Pdust* deposition are close to the inter-annual average for the 1997-2012 period. »

[Figure]

*Figure 2: Maps of average deposition flux (top), standard deviation (middle), and standard deviation to average ratio (bottom) of Pcomb deposition over the entire LMDz-INCA domain. Average over the 1997-2012 period*

[Figure]

*Figure 3: Same as Figure 2, zoomed over the Mediterranean Sea*

*Pg 17 line 559: "The relative effects of each source are maximal in their areas of maximal deposition and can induce an enhancement of up to 30 % in biological productivity during the period of surface water stratification."*

*In my opinion Authors should specify that the 30 % increase in biological productivity refers to the 1-10m surface layer, and it doesn't refer to the total water column vertically integrated primary productivity. Moreover during summer a consistent part of the productivity could be located in the subsurface layers, below 10 m depth, therefore 30% could have lower relative impact considering the vertically integrated PP.*

Reviewer is right, we added this precision.

*Minor comment*
*Pg 17 line 578. 85% of of P*

Changed

**Responses to editor's comments**

*Abstract: Lines 10-11: Please clarify how this conclusion holds for the other years.*
We added the sentence : « The evaluation of monthly averaged deposition fluxes variability of Pdust and Pcomb for the 1997-2012 period indicates that these conclusion may hold true for different years. »

*Introduction, line 31-33: '… for marine biology..'' I would be more specific.*
Changed to « for marine primary productivity »

*Line 42: What do you mean by "constrain"?*
Changed to « to better characterize »

*Line 64: ". The Mediterranean Sea is also a hot–spot for climate change impacts (Lejeusne et al., 2010)" This sentence seems to be a little bit disconnected from the rest of the paragraph. I suggest that you either remove it or that you detail somehow the link between climate change and aerosols emission.*
Sentence removed

*Lines 67-69: please give a reference*
Reference to Peñuelas *et al.* 2013 added.

*Lines 76-80: Please synthetize in 2-3 lines the main outcome of Richon et al 2017*
Sentence added :  « Their results showed important impacts of N deposition on biological productivity (primary production, chlorophyll *a* production, plankton and bacterial concentrations) in the northern Ionian and Levantine basins and limited, yet significant impact of P deposition in the southern Mediterranean regions. »

*Line 81: I would say: "..;by further considering the contribution of P from combustion sources in addition to that from anthropogenic…"*
Changed

*Line 101: "the authors" please specify who? Hamon et al ., 2015*
Changed to Hamon et al.

*Line 113: I would not say that PISCES is a Monod-type model but rather that the uptake of nutrient (e.g; nitrate, ammonium, phosphate, iron and silicate) by phytoplankton is governed by a Monod-type model. Besides, I would start by a general description of PISCES (e.g. lines 118-122) then I would go to specific feature of the model (e.g. description of nutrient uptake formulation).*
This part was reorganized.

*Line 118: trophic level and not biological levels.*
Changed

*Line 135: "Biogeochemical variables are prognostically calculated instead of being read from forcing files". This sentence is not necessary (you have a model so the model computes the state variables.*
Removed

*Line 136: "Biogeochemical characteristics of the latest version of the NEMOMED12/PISCES model are evaluated in Richon et al. (2017). » This sentence is similar to the next one. Please uniformize the definition of chlorophyll a. Sometimes you use Chl a, sometimes chlorophyll a.*
Sentence removed and we write « chlorophyll *a »*

*Lines 190-191: "The Pcomb deposition fields from LMDz–INCA used here have a coarser resolution than for Pdust of 1.27◦ in latitude by 2.5◦ in longitude". This is confusing since we are told that the study wants to have consistent atmospheric inputs provided by the same model (see lines 162-164). Please clarify.*
The *Pcomb* deposition fields used here were obtained from a different simulation performed with the same model LMDz-INCA on a coarser grid resolution. We added these precisions in the text.

*Line201: ".. which leads to a general agreement of modeled surface P concentrations with …" to which model are you referring? I guess that this is a global estimation? Please specify.*
We precised « general agreement of their globally modeled surface P concentrations... »

*Lines 207-208: P surface concentrations and deposition produced by LMD model? If yes, please link appropriately with Lines 209- which refer to LMDz performances as well because as it is written we have the feeling that these two parts are not related to the same model evaluation.*
These lines refer to LMDz-INCA. We added the precision.

*Line 255: something is missing in the sentence. Please reformulate.*
The correct sentence is : « In Table 2, the dust deposition fluxes for the period 2001-2002, corresponding to the ADIOS campaign, are based on model outputs with a lower resolution (1.27°in latitude by 2.5°in longitude) than those for the year 2005 (0.94°in latitude by 1.28°in longitude). »

*Line 310: ".. available at a too coarse time resolution to perform oceanic simulations..". Please justify because looking at figure 3 we do not have the feeling that Pcomb has significant variability.*
Changed to « for which monthly depositon fluxes are available ». We did not run NEMOMED12/PISCES for 16 years with monthly P deposition, but instead, we provide in Figures 2 and 3 the time series of *Pcomb* deposition over the 1997-2012 period and the map of standard deviation to average ratio for the same time period. These figures show that *Pcomb* deposition has low variability, and that 2005 can be considered a suitable study year.

*Line 331: this is clearer to use Table 3 instead of This table.*
*Changed*

*Line 344: You are not really assessing the impact of atmospheric deposition on the phosphate budget but rather on the phosphate surface concentration. So I would remove budget.*
Changed

*Line 446: Please define PM2.5 and PM 10.I do not think that it has been already defined above.*
Done

*Lines 451-454: "Although total mass deposition of phosphorus from desert dust exceeds that of combustion aerosols, the latter are much more soluble than lithogenic dust. » Please clarify. In the abstract we are told the reverse. Do you mean that the higher flux in the abstract results from the use of different solubility coefficients?*
In the abstract, we give soluble phosphate deposition fluxes. The term « phosphorus » should be replaced by « particulate phosphorus ».

*Legend figure 5: "..; map represents the 50 % Pdust proportion limit" compared to 2005 for Pcomb? Or 1997-2012 for Pcomb as well? Please clarify.*
1997-2012

*Figure 9: legend: I guess that it is the reference primary production and not the reference PO4.*
Yes

*Table 3: "Relative atmospheric contribution (%) to total PO4 supply in different sub–basins of atmospheric sources". I would remove "of atmospheric sources"?*
Removed